# Illuminating the Plant Rhabdovirus Landscape through Metatranscriptomics Data

**DOI:** 10.3390/v13071304

**Published:** 2021-07-05

**Authors:** Nicolás Bejerman, Ralf G. Dietzgen, Humberto Debat

**Affiliations:** 1Instituto de Patología Vegetal, Centro de Investigaciones Agropecuarias, Instituto Nacional de Tecnología Agropecuaria (IPAVE—CIAP—INTA), Camino 60 Cuadras Km 5.5, Córdoba X5020ICA, Argentina; debat.humberto@inta.gob.ar; 2Consejo Nacional de Investigaciones Científicas y Técnicas, Unidad de Fitopatología y Modelización Agrícola, Camino 60 Cuadras Km 5.5, Córdoba X5020ICA, Argentina; 3Queensland Alliance for Agriculture and Food Innovation, The University of Queensland, St. Lucia, Brisbane, QLD 4072, Australia; r.dietzgen@uq.edu.au

**Keywords:** plant rhabdovirus, evolution, taxonomy, metatranscriptomics

## Abstract

Rhabdoviruses infect a large number of plant species and cause significant crop diseases. They have a negative-sense, single-stranded unsegmented or bisegmented RNA genome. The number of plant-associated rhabdovirid sequences has grown in the last few years in concert with the extensive use of high-throughput sequencing platforms. Here, we report the discovery of 27 novel rhabdovirus genomes associated with 25 different host plant species and one insect, which were hidden in public databases. These viral sequences were identified through homology searches in more than 3000 plant and insect transcriptomes from the National Center for Biotechnology Information (NCBI) Sequence Read Archive (SRA) using known plant rhabdovirus sequences as the query. The identification, assembly and curation of raw SRA reads resulted in sixteen viral genome sequences with full-length coding regions and ten partial genomes. Highlights of the obtained sequences include viruses with unique and novel genome organizations among known plant rhabdoviruses. Phylogenetic analysis showed that thirteen of the novel viruses were related to cytorhabdoviruses, one to alphanucleorhabdoviruses, five to betanucleorhabdoviruses, one to dichorhaviruses and seven to varicosaviruses. These findings resulted in the most complete phylogeny of plant rhabdoviruses to date and shed new light on the phylogenetic relationships and evolutionary landscape of this group of plant viruses. Furthermore, this study provided additional evidence for the complexity and diversity of plant rhabdovirus genomes and demonstrated that analyzing SRA public data provides an invaluable tool to accelerate virus discovery, gain evolutionary insights and refine virus taxonomy.

## 1. Introduction

The costs for high-throughput sequencing (HTS) have been significantly reduced each year due to advances in sequencing technologies; therefore, the number of genome and transcriptome sequencing projects has been steadily increasing, resulting in a massive number of nucleotides deposited in the Sequence Read Archive (SRA) of the National Center for Biotechnology Information (NCBI). Over 16,000 petabases (10^15^ bases) have been deposited in the SRA, with over 6000 petabases available as open-access data [1]. Thus, this large amount of data has provided significant challenges for data storage, bioinformatic analysis and management. This impressive and potentially useful amount of data concomitantly raised two issues: (I) high logistical costs of data management and (II) large amounts of neglected and unused data awaiting secondary analysis and repurposing. In the specific case of large plant sequencing project datasets, virome studies are scarce.

Abundant novel viruses, many of them not known to induce any apparent symptoms in their host or without a known host, have been identified from diverse environments using metagenomic approaches. This has highlighted our limited knowledge about the richness of a continuously expanding plant virosphere, which appears highly diverse in every potential host assessed so far [2,3,4,5]. Furthermore, the great number of viruses recently discovered by HTS, a miniscule portion of the virosphere, allowed a first glimpse of the path to a comprehensive megataxonomy of the virus world [6].

The scientific interest of the submitters of transcriptome datasets is often limited to a narrow objective within their specific field of study, which leaves a large amount of potentially valuable data not analyzed [7]. In such transcriptome datasets, viral sequences may be hidden in plain sight; thus, their analysis has become a valuable tool for the discovery of novel viral sequences [8,9,10,11,12,13,14,15,16]. In a recent consensus statement report, Simmonds and colleagues [17] contended that viruses that are known only from metagenomic data can, should and have been incorporated into the official classification scheme overseen by the International Committee on Taxonomy of Viruses (ICTV). Consequently, the analysis of public sequence databases constitutes a valuable resource for the discovery of novel plant viruses, which allows the reliable identification and characterization of new viruses in hosts with no previous record of virus infections [8]. This approach to virus discovery is inexpensive, as it does not require the acquisition of samples and subsequent sequencing but on secondary analyses of publicly available data to address novel research questions and objectives. At the same time, it is more wide-ranging and comprehensive than any other current approach due to the millions of datasets from a large variety of potential host species available from the NCBI-SRA [12].

Plant rhabdoviruses have negative-sense, single-stranded RNA genomes and are taxonomically classified in six genera: *Cytorhabdovirus*, *Alphanucleorhabdovirus*, *Betanucleorhabdovirus* and *Gammanucleorhabdovirus* for viruses that have an unsegmented genome and *Dichorhavirus* and *Varicosavirus* for viruses that have a bisegmented genome and infect both monocot and dicot plants [18]. These six genera were recently assigned to the subfamily *Betarhabdovirinae* within the family *Rhabdoviridae* [19]. Viruses classified in five of these genera are transmitted persistently by arthropods in which they also replicate [18,20], whereas varicosaviruses are transmitted by soil-borne chytrid fungi [18]. Cyto- and nucleorhabdovirus genomes have six conserved canonical genes encoding in the order 3′-nucleocapsid protein (N)-phosphoprotein (P)–putative movement protein (P3)-matrix protein (M)-glycoprotein (G)–large polymerase (L)-5′; the L gene of dichorhaviruses is located on RNA2 [21]. Up to three accessory genes with unknown functions have been identified among cyto- and nucleorhabdovirus genomes, leading to diverse genome organizations [21,22]. Conserved gene junction sequences separate each gene, and the overall coding region is flanked by 3′ leader and 5′ trailer sequences that feature partially complementary ends that may form a panhandle structure during replication [20]. Varicosavirus RNA 1 has one to two genes, with one of those encoding the RNA-dependent RNA polymerase L, while RNA 2 has three to five genes, with the first open reading frame (ORF) encoding a coat protein [20,21]. The 3′- and 5′-terminal sequences of the two varicosavirus genome segments are similar but do not exhibit inverse complementarities [21].

In this study, we queried the publicly available plant transcriptome datasets in the transcriptome shotgun assembly (TSA) database hosted at NCBI and identified 27 novel plant rhabdoviruses from 25 plant and one insect species, showing structural, functional and evolutionary cues to be classified in the family *Rhabdoviridae*; subfamily *Betarhabdovirinae* and genera *Cytorhabdovirus*, *Alphanucleorhabdovirus*, *Betanucleorhabdovirus, Dichorhavirus* and *Varicosavirus*.

## 2. Materials and Methods

### 2.1. Identification of Plant Rhabdovirus Sequences from Public Plant Transcriptome Datasets

The detection of plant rhabdovirus sequences was done as described by Longdon and colleagues [13]. Briefly, the amino acid sequences corresponding to the nucleocapsid and polymerase proteins of several known cyto- and nucleorhabdoviruses were used as query in tBlastn searches with the parameters word size = 6, expected threshold = 10 and scoring matrix = BLOSUM62 against the Viridiplantae (taxid:33090) and Hemiptera (taxid:7524) TSA databases. The obtained hits were explored by eye and based on the percentage identity, query coverage and E-value (>1 × 10^5^), shortlisted as likely corresponding to novel virus transcripts, which were then further analyzed. Given the redundant nature of many retrieved hits, a step of contig clustering was implemented using the CD-hit suite with the standard parameters available at http://weizhongli-lab.org/cdhit_suite/cgi-bin/index.cgi?cmd=cd-hit (accessed on 10 March 2021). In addition, the raw sequence data corresponding to several SRA experiments associated with different NCBI Bioprojects (Table 1) were retrieved for further analyse

### 2.2. Sequence Assembly and Identification

The nucleotide (nt) raw sequence reads from each analyzed SRA experiment linked to the TSA projects returning rhabdovirus-like hits were downloaded and preprocessed by trimming and filtering with the Trimmomatic tool, as implemented in http://www.usadellab.org/cms/?page=trimmomatic (accessed on 15 March 2021), and the resulting reads were assembled de novo with Trinity v2.6.6 using the standard parameters. The transcripts obtained from de novo transcriptome assembly were subjected to bulk local BLASTX searches (E-value < 1 × 10^5^) against a Refseq virus protein database available at ftp://ftp.ncbi.nlm.nih.gov/refseq/release/viral/viral.1.protein.faa.gz (accessed on 3 March 2021). The resulting viral sequence hits of each bioproject were visually explored. Tentative virus contigs were extended by iterative mapping of each SRA library’s raw reads. This strategy employed BLAST/nhmmer to extract a subset of reads related to the query contig, and these retrieved reads were used to extend the contig, and then, the process was repeated iteratively using as the query the extended sequence. The extended and polished transcripts, now presenting overlapping regions, were reassembled using the Geneious v8.1.9 (Biomatters Ltd., Auckland, New Zealand) alignment tool.

### 2.3. Bioinformatics Tools and Analyses

#### 2.3.1. Sequence Analyses

ORFs were predicted with ORFfinder (https://www.ncbi.nlm.nih.gov/orffinder/) (accessed on 19 March 2021), functional domains, and the architecture of the translated gene products was determined using InterPro (https://www.ebi.ac.uk/interpro/search/sequence-search) (accessed on 19 March 2021) and the NCBI Conserved domain database v3.16 (https://www.ncbi.nlm.nih.gov/Structure/cdd/wrpsb.cgi) (accessed on 19 March 2021). Further, HHPred and HHBlits as implemented in https://toolkit.tuebingen.mpg.de/#/tools/ (accessed on 19 March 2021) were used to complement the annotation of divergent predicted proteins by hidden Markov models. Importin-α-dependent nuclear localization signals were predicted using cNLS Mapper available at http://nls-mapper.iab.keio.ac.jp/ (accessed on 23 March 2021), nuclear export signals were predicted using NetNES 1.1 available at www.cbs.dtu.dk/services/NetNES/ (accessed on 19 March 2021) and transmembrane domains were predicted using the TMHMM version 2.0 tool (http://www.cbs.dtu.dk/services/TMHMM/) (accessed on 23 March 2021).

#### 2.3.2. Pairwise Sequence Identity

Percentage amino acid (aa) sequence identities of the predicted ORFs of each plant-associated rhabdovirid identified in this study based on the available plant-associated rhabdovirus genome sequences were calculated using https://www.ebi.ac.uk/Tools/psa/emboss_needle/ (accessed on 23 March 2021).

#### 2.3.3. Phylogenetic Analysis

Phylogenetic analysis based on the predicted nucleocapsid proteins of all plant rhabdovirids, listed in Appendix A, was done using MAFFT 7 https://mafft.cbrc.jp/alignment/software (accessed on 29 March 2021), with multiple aa sequence alignments using FFT-NS-i as the best-fit model. The aligned aa sequences were used as the input in MegaX software [23] to generate phylogenetic trees by the maximum-likelihood method (best-fit model = WAG + G + F). Local support values were computed using bootstraps with 1000 replicates.

**Table 1 viruses-13-01304-t001:** Summary of rhabdovirus contigs identified from plant transcriptome data available in the National Center for Biotechnology Information (NCBI) database.

Genus	Plant/Insect Host; Monocot/Dicot	Virus Name	Abbreviation	Bioproject ID	Data Citation	Segment Number	Length (nt)	Accession Number	Protein ID	Length (aa)
*Alphanucleorhabdovirus*	Blue agave (*Agave tequilana*); monocot	*Agave tequilana* virus 1	ATV1	PRJNA193469	[24]	1	13,166	BK014297	N P P3 M G L	452 335 287 257 597 1937
*Betanucleorhabdovirus*	Common milkweed (*Asclepias syriaca*); dicot	*Asclepias syriaca* virus 2	AscSyV2	PRJNA210776	[25]	1	12,940	BK014299	N P P3 M G L	459 337 343 250 643 2023
	Giant dodder (*Cuscuta reflexa*); dicot	*Cuscuta reflexa* virus 1	CusReV1	PRJNA290291	[26]	1	8869 *	BK014340; BK014341; BK014342; BK014343; BK014344; BK014345	N P P3 M G L	457 336 325 269 * 550 * 740 *
	Small water-pepper (*Persicaria minor*); dicot	*Persicaria minor* virus 1	PerMiV1	PRJNA208436	[27]	1	4793 *	BK014346; BK014347; BK014348; BK014349; BK014350; BK014351; BK014352; BK014353; BK014354	N P P3 M G L	372 * 115 * 172 * 227 * 260 * 430 *
	Cubanoregano (*Plectranthus aromaticus*); dicot	*Plectranthus aromaticus* virus 1	PleArV1	PRJNA491230	Ab Rahim, M.H.; University Malaysia Pahang, Malaysia; unpublished	1	12,994	BK014300	N P P3 M G L	450 332 321 286 582 2085
	Red rhododendron (*Rhododendron delavayi*); dicot	*Rhododendron delavayi* virus 1	RhoDeV1	PRJNA358123	[28]	1	13,719	BK014301	N P P3 M G L	464 338 326 279 636 2016
*Cytorhabdovirus*	Chinese onion (*Allium chinense*); monocot	*Allium chinense* virus 1	AChV1	PRJNA310810	[29]	1	7981 *	BK014319; BK014320; BK014321; BK014322; BK014323; BK014324	N P P′ P3 M G P6 L	457 316 128 231 165 542 * - 531*
	Flamingo lily (*Anthurium amnicola*); monocot	*Anthurium amnícola* virus 1	AntAmV1	PRJNA288827	[30]	1	12,480	BK014302	N P P3 M G L	409 314 207 176 565 2047
	*A. syriaca*	*Asclepias syriaca* virus 1	AscSyV1	PRJNA210776	[25]	1	13,392	BK014298	N P P′ P3 M G P6 L	445 304 87 334 176 551 86 2101
	Silverleaf whitefly *(Bemisia tabaci*)	*Bemisia tabaci* -associated virus 1	BeTaV1	PRJNA237273	[31]	1	13,025	BK014303	N P P3 P4 M G L	447 326 187 48 206 579 2102
	Yam (*Dioscorea composita*)	*Dioscorea composita* virus 1	DiCoV1	PRJNA253902	[32]	1	9959 *	BK014355; BK014356; BK014357; BK014358	N P P3 M G P6 L	441 N/A N/A 201 586 72 1546 *
	Beach silvertop (*Glehnia littoralis*); dicot	*Glehnia littoralis* virus 1	GlLV1	PRJNA248158	[33]	1	12,193	BK014304	N P P′ P3 M G P6 L	412 324 84 201 167 551 66 2072
	Marsh fragrant orchid (*Gymnadenia densiflora*); monocot	*Gymnadenia densiflora* virus 1	GymDenV1	PRJNA504609	[34]	1	9887	BK014305	N P M L	444 310 189 2068
	Bird’s foot trefoil (*Lotus corniculatus*); dicot	*Lotus corniculatus* virus 1	LotCorV1	PRJNA77207	[35]	1	12,599 *	BK014306	N P P′ P3 M G P6 L	479 311 98 362 179 558 56 1985 *
	European white waterlily (*Nymphaea alba*); dicot	*Nymphaea alba* virus 1	NymAV1	PRJNA472003	Unlu E.S., and Yildiz, G.G; Abant Izzet Baysal University, Turkey; unpublished	1	12,886	BK014307	N P P′ P3 M G P6 L	421 308 85 228 195 560 69 2076
	Crowfoot geranium (*Pelargonium radula*), dicot	*Pelargonium radula* virus 1	PelRaV1	PRJNA491235	Ab Rahim, M.H.; University Malaysia Pahang, Malaysia; unpublished	1	11,130 *	BK014325; BK014326; BK014327; BK014328	N P P′ P3 M G P6 L	483 309 69 354 177 578 * 54 1391*
*Cytorhabdovirus*	Seepweed (*Suaeda salsa*); dicot	*Suaeda salsa* virus 1	SuSV1	PRJNA395283	[36]	1	6156 *	BK014359; BK014360; BK014361; BK014362; BK014363; BK014364; BK014365; BK014366	N P P′ P3 M G P6 L	332 * 294 * 66 * 218 165 70 * - 711*
	African marigold (*Tagetes erecta*); dicot	*Tagetes erecta* virus 1	TaEV1	PRJNA431782	[37]	1	11,707	BK014308	N P P3 M L	520 505 200 187 2079
	Ajwain (*Trachyspermum ammi*); dicot	*Trachyspermum ammi* virus 1	TrAV1	PRJNA359623	[38]	1	10,920	BK014309	N P P3 M L	455 343 237 200 2069
*Dichorhavirus*	Hidden violet (*Viola verecunda*); dicot	*Viola verecunda* virus 1	VVeV1	PRJNA345302	[39]	1 2	5304 * 5212 *	BK014329; BK014330; BK014331 BK014332; BK014333	N P P3 M G L	494 N/A 323 N/A 661 1668 *
*Varicosavirus*	Mouse garlic (*Allium angulosum*); monocot	*Allium angulosum virus 1*	AAnV1	PRJNA542932	[40]	1 2	6679 4560	BK059208 BK059209	L N 2 3	2048 481 454 129
	Bok choy (*Brassica rapa subsp. chinensis*); dicot	*Brassica rapa* virus 1	BrRV1	PRJNA396268	[41]	1 2	6397 4068	BK014310 BK014311	L N 2 3	2019 435 411 180
	Perennial ryegrass (*Lolium perenne*); monocot	*Lolium perenne* virus 1	LoPV1	PRJNA222646	[42]	1 2	6302 4167	BK014312 BK014313	L N 2 3	2029 533 379 161
	Asian cow-wheat (*Melampyrum roseum*); dicot	*Melampyrum roseum* virus 1	MelRoV1	PRJDB5395	[43]	1 2	6365 4635	BK014314 BK014315	L N 2 3 4	2003 443 350 304 193
	Downy phlox (*Phlox pilosa*); dicot	*Phlox pilosa* virus 1	PhPiV1	PRJNA360978	[44]	1 2	4896 * 3201 *	BK014334; BK014335; BK014336; BK014337 BK014338; BK014339	L N 2 3 4	1589 * 378 81 * 289 110*
	Limber pine (*Pinus flexilis*); gymnosperm	*Pinus flexilis* virus 1	PiFleV1	PRJNA315892	[45]	1	11,740	BK014316	N 2 3 4 L	405 447 318 219 2049
	Spinach (*Spinacia oleracea*); dicot	spinach virus 1	SpV1	PRJDB3392	[46]	1 2	6151 * 3750 *	BK014317 BK014318	L N 2 3	2010 * 439 450 136*

* Partial sequence; N/A: not available.

## 3. Results

### 3.1. Summary of Discovered Rhabdovirid Sequences

The complete coding regions of 17 novel rhabdoviruses were identified; in addition, partial genomic sequences for 10 novel viruses were assembled. These viruses were associated with 25 plant host species and one insect species (Table 1). The bioinformatic and source data of each of the 27 viral sequences, as well as the GenBank accession number and proposed classification, are listed in Table 1; the summary of the assembly statistics of each virus of the plant rhabdovirus sequences identified from the transcriptome data available in the NCBI database are presented in Appendix A. Based on phylogenetic relatedness, genome organization and sequence identity, the novel viruses were tentative assigned to the established plant rhabdovirus genera *Alphanucleorhabdovirus*, *Betanucleorhabdovirus*, *Cytorhabdovirus*, *Dichoravirus* and *Varicosavirus*. Most of the tentative plant hosts of the novel viruses are herbaceous dicots (16/25), seven are herbaceous monocots, one a woody dicot and one a gymnosperm (Table 1). The characteristics of deduced proteins encoded by each rhabdovirid sequence were determined by predictive algorithms and are shown in Appendix A. The genomic architecture and evolutionary placement of the 27 discovered viruses are described below, grouped by affinity to members of the diverse genera within the *Betarhabdovirinae*.

### 3.2. Alphanucleorhabdovirus

The complete coding region of a novel putative alphanucleorhabdovirus, tentatively named *Agave tequilana* virus 1 (ATV1), with the genome organization 3′-N-P–P3-M-G-L-5′ (Figure 1A) was assembled from blue agave transcriptome data (Table 1). A nuclear localization signal (NLS) was predicted in every ATV1-encoded protein (Appendix A). According to the NLS scores, the N protein is predicted to be located exclusively in the nucleus, whereas the P and L proteins have a partial nuclear localization, while the P3, M and G proteins are localized to both the nucleus and the cytoplasm. Leucine-rich nuclear export signals (NES) were predicted in the N, P and L proteins (Appendix A). A transmembrane domain motif was detected in the C-terminus of the G protein, and a signal peptide was predicted in its N-terminus (Appendix A). The consensus gene junction sequence 3′-AUUCUUUUUGGGUUG-5′ of the ATV1 genome is similar to that of the alphanucleorhabdoviruses maize mosaic virus (MMV), maize Iranian mosaic virus (MIMV), Morogoro maize-associated virus (MMaV) and taro vein chlorosis virus (TaVCV) (Table 2).

Pairwise aa sequence identities between the ATV1-encoded proteins and those from other alphanucleorhabdoviruses showed low sequence identities of 10.9–35.0% (Table 3). The nucleotide sequence identity for the complete genome sequence of ATV1 and other alphanucleorhabdoviruses ranged from 47–49.3 % (Table 3).

The phylogenetic analysis based on the N protein aa sequences showed that ATV1 clustered with the monocot-infecting alphanucleorhabdoviruses MMV, MIMV, MMaV and TaVCV (Figure 2), which were also the most similar in pairwise sequence identity values, shared equivalent genome organizations and had similar conserved gene junction sequences (Table 2).

### 3.3. Betanucleorhabdoviruses

The complete coding regions of three putative betanucleorhabdoviruses, tentatively named *Asclepias syriaca* virus 2 (AscSyV2), *Plectranthus aromaticus* virus 1 (PleArV1) and *Rhododendron delavayi* virus 1 (RhoDeV1), as well as the partial genomes of *Cuscuta reflexa* virus 1 (CusReV1) and *Persicaria minor* virus 1 (PerMiV1), were assembled in this study (Table 1). The genome organization of these five viruses is 3′-N-P–P3-M-G-L-5′ (Figure 1A). A NLS was predicted in every protein encoded by AscSyV2, CusReV1, PleArV1 and RhoDeV1 (Appendix A). According to the NLS scores, the CusReV1 and PleArV1 N proteins, RhoDeV1 P protein, CusReV1 M protein and PleArV1 and RhoDeV1 L proteins are expected to be exclusively nuclear, whereas the RhoDeV1 M protein is predicted to have a partial nuclear localization. The AscSyV2 and RhoDeV1 N proteins; AscSyV2, CusReV1 and PleArV1 P proteins; AscSyV2, CusReV1, PleArV1 and RhoDeV1 P3 proteins; AscSyV2 and PleArV1 M proteins; AscSyV2, CusReV1, PleArV1 and RhoDeV1 G proteins and AscSyV2 and CusReV1 L proteins are predicted to localize to both the nucleus and the cytoplasm. Leucine-rich NES were predicted in the AscSyV2, CusReV1 and PleArV1 N proteins; the CusReV1 and PleArV1 P proteins; the CusReV1 and RhoDeV1 P3 proteins; the PleArV1 M protein and the PleArV1 and RhoDeV1 L proteins (Appendix A). A transmembrane domain was detected in the C-terminal sequence of the AscSyV2, CusReV1, PleArV1 and RhoDeV1 G proteins. A signal peptide was detected in the G protein N-terminus of AscSyV2, CusReV1 and RhoDeV1 (Appendix A).

The deduced AscSyV2 M protein, PleArV1 G protein and RhoDeV1 L protein are the smallest such proteins reported so far among betanucleorhabdoviruses (Appendix A).

Interestingly, the N proteins of AscSyV2, PleArV1 and RhoDeV1 are basic, whereas the CusReV1 N protein is acidic. Moreover, the CusReV1 and PleArV1 M proteins are basic, while the AscSyV2 and RhoDeV1 M proteins are acidic (Appendix A).

The consensus gene junction sequence 3′-AUUCUUUUU GG UUG-5′ of all five novel viral genomes is identical and the same as in the genome of every betanucleorhabdovirus described thus far: alfalfa-associated nucleorhabdovirus (AaNV), apple rootstock virus A (ApRVA), *Bacopa monnieri* virus 2 (BmV2), bird’s-foot trefoil associated virus 1 (BFTaV1), black currant-associated rhabdovirus (BCaRV), cardamom vein clearing virus 1 (CdVCV1), datura yellow vein virus (DYVV), green Sichuan pepper nucleorhabdovirus (GSPNuV), sonchus yellow net virus (SYNV) and sowthistle yellow vein virus (SYVV) (Table 2).

Pairwise aa sequence identity values between each encoded protein of the five novel viruses and those from other betanucleorhabdoviruses vary significantly (Table 4). The highest nucleotide sequence identity for the genome sequence between AscSyV2, PleArV1 and RhoDeV1 and other betanucleorhabdoviruses ranged from 51.4 to 62.1% (Table 4).

The phylogenetic analysis based on the deduced N protein aa sequences showed that CusReV1, RhoDeV1, PerMiV1 and PleArV1 clustered with the betanucleorhabdoviruses BFTaV1, BCaRV, BmV2, CdVCV1, DYVV, GSPNuV, SYNV and SYVV (Figure 2), which are also the most similar in pairwise sequence identity values of their cognate proteins; furthermore, all these viruses have a similar genome organization. Interestingly, AscSyV2 clustered closely with the betanucleorhabdoviruses AaNV and ApRVA and had similar pairwise sequence identity values for their cognate proteins. However, the genome organization of AaNV and ApRVA was different in that it included an accessory gene between the M and G genes.

### 3.4. Cytorhabdoviruses

The complete coding region of eight novel cytorhabdoviruses, tentatively named *Anthurium amnicola* virus 1 (AntAmV1), *Asclepias syriaca* virus 1 (AscSyV1), *Bemisia tabaci*-associated virus 1 (BeTaV1), *Glehnia littoralis* virus 1 (GlLV1), *Gymnadenia densiflora* virus 1 (GymDenV1), *Nymphaea alba* virus 1 (NymAV1), *Tagetes erecta* virus 1 (TaEV1) and *Trachyspermum ammi* virus 1 (TrAV1), as well as the partial genomes of five other cytorhabdoviruses, named *Allium chinense* virus 1 (AChV1), *Dioscorea composite* virus 1 (DiCoV1), *Lotus corniculatus* virus 1 (LotCorV1), *Pelargonium radula* virus 1 (PelRaV1) and *Suaeda salsa* virus 1 (SuSV1), were assembled in this study (Table 1). The genome organization of these thirteen novel cytorhabdoviruses is highly diverse. AntAmnV1 and likely DiCoV1 lack additional accessory genes and have the genome organization 3′-N-P–P3-M-G-L-5′ (Figure 1B). AscSyV1, GlLV1, LotCorV1, NymAV1, PelRaV1 and, likely, AChV1 and SuSV1 have an accessory ORF between the G and L genes, with the genome organization 3′-N-P–P3-M-G-P6-L-5′ (Figure 1B), while BeTaV1 have an accessory ORF between the P3 and M genes, with the genome organization 3′-N-P–P3-P4-M-G-L-5′ (Figure 1B). The genome organization of GmyDenV1, TaEV1 and TrAV1 differs from that of all other known plant rhabdoviruses: the GymDenV1 genome appears to only have four genes in the order 3′-N-P–M-L-5′ (Figure 1B), while TaEV1 and TrAV1 genomes have five genes in the order 3′-N-P–P3-M-L-5′ but do not encode a glycoprotein (Figure 1B). An overlapping ORF within the P-encoding ORF, named P′, is present in the AChV1, AscSyV1, GilV1, LotCorV1, NymAV1, PelRaV1 and SuSV1 genomes (Figure 1B and Table 1). The predicted TaEV1 N protein is the largest described so far among the cytorhabdoviruses, whereas the AntAmV1 L protein is the smallest reported so far among the cytorhabdoviruses (Appendix A).

Interestingly, the predicted N protein of almost every cytorhabdovirus identified in our study is acidic, but the PelRaV1 N protein is neutral, and the BeTaV1 N protein is basic. The P protein of almost every cytorhabdovirus identified in our study is acidic, but the TrAV1 P protein is basic. Moreover, the M protein of almost every cytorhabdovirus identified in our study is basic, except GymDenV1 and TrAV1 M proteins, which are acidic. Furthermore, the predicted G proteins of AscSyV1, DiCoV1, GlLV1, NymAV1 and PelRaV1 are acidic, except the LotCorV1 G protein is neutral, and the AntAmV1 and BeTaV1 G proteins are basic (Appendix A).

A putative transmembrane domain was identified in the G protein C-terminal sequence of AChV1, AntAmV1, AscSyV1, BeTaV1, DiCoV1, GlLV1, LotCorV1, NymAV1 and PelRaV1; two additional transmembrane domains were identified in the G protein N-terminus of AntAmV1 and BeTaV1 and another one in the N-terminus of PelRaV1 G protein. A transmembrane domain was also predicted in every P6 protein of these viruses (Appendix A). Moreover, one transmembrane domain was predicted in the SuSV1 P′ protein and two in the AChV1, GlLV1, LotCorV1 and PelRaV1 P′ proteins, while three were predicted in the AscSyV1 and NymAV1 P′ proteins (Appendix A). A signal peptide sequence was identified in the G protein N-terminus of AChV1, AscSyV1, GlLV1, LotCorV1, NymAV1 and PelRaV1 (Appendix A). Interestingly, a leucine-rich NES was predicted in the AChV1, AntAmV1, AscSyV1, BeTaV1, DiCoV1, GlLV1, GymDenV1, LotCorV1, PelRaV1, SuSV1, TaEV1 and TrAV1 N proteins; the BeTaV1, GymDenV1, LotCorV1, NymAV1, PelRaV1, SuSV1 and TrAV1 P proteins; the AChV1, AntAmV1, AscSyV1, DiCoV1, GymDenV1, LotCorV1, NymAV1, PelRaV1, SuSV1 and TrAV1 M proteins and the PleArV1 and RhoDeV1 L proteins (Appendix A).

The consensus gene junction sequences of the novel cytorhabdoviruses identified in our study are diverse (Table 2) but similar to those previously reported for phylogenetically related cytorhabdoviruses. Interestingly, the intergenic sequence of the GymDenV1 and TrAV1 gene junctions starts with an adenine instead of a guanine residue like in every other plant rhabdovirus (Table 2).

Pairwise aa sequence identity values between each encoded protein of the thirteen novel viruses and those from known cytorhabdoviruses vary significantly (Table 5). The highest nucleotide sequence identity for the genome sequence between AntAmV1, AscSyV1, BeTaV1, GlLV1, GymDenV1, NymAV1, TaEV1 and TrAV1 and the other cytorhabdoviruses ranged between 50.3 and 65.6% (Table 5).

The phylogenetic analysis based on the N protein aa sequence showed that the thirteen novel viruses grouped with known cytorhabdoviruses GlLV1 and NymAV1 clustered with Kenyan potato cytorhabdovirus (KePCyV), *Trifolium pratense* virus A (TpVA), strawberry virus 1 (StrV1) and tomato yellow mottle-associated virus (TYMaV), with whom they share a similar genomic organization. AChV1 and SuSV1 formed a clade with alfalfa dwarf virus (ADV), chrysanthemum yellow dwarf-associated virus (ChYDaV), raspberry vein chlorosis virus (RVCV) and strawberry crinkle virus (SCV). AscSyV1, LotCorV1 and PelRaV1 clustered with Actinidia cytorhabdovirus (AcCV), *Bacopa monnieri* virus 1 (BmV1) and Wuhan insect virus 4 (WhIV4), and all these viruses have a similar genomic organization. BeTaV1 clustered with cucurbit cytorhabdovirus 1 (CuCV1), and these viruses share a similar genomic organization. AntAmV1 clustered with paper mulberry mosaic-associated virus (PMuMaV), with which it shares a similar genomic organization. TaEV1 clustered with the cluster formed by AntAmV1 and PMuMaV, but these viruses have a dissimilar genomic organization. DiCoV1 clustered with Colocasia bobone disease-associated virus (CBDaV), but these viruses have a dissimilar genomic organization, while GymDenV1 and TrAV1 formed a monophyletic sister clade with the aphid-transmitted cytorhabdoviruses (Figure 2).

### 3.5. Dichorha-Like Viruses

The near-complete RNA 1 and RNA 2 sequences of a dichorha-like virus tentatively named *Viola verecunda* virus 1 (VVeV1) were identified in this study. We assembled three nonoverlapping fragments corresponding to RNA 1 containing the complete N, P3 and G genes and the near-complete RNA 2, which encodes the L protein (Table 1). Thus, the likely genome organization of VVeV1 is 3′-N-P-P3-M-G-5′ for RNA 1 and 3′-L-5′ for RNA 2 (Figure 1D). A NLS was predicted in every deduced VVe1 protein (Appendix A). According to the NLS scores, the N and L proteins should be exclusively localized to the nucleus, whereas the P3 and G proteins are predicted to localize to both the nucleus and the cytoplasm. Leucine-rich NESs were predicted in the N and L protein sequences (Appendix A). A transmembrane domain was identified in the C-terminus of the G protein (Appendix A). We were not able to identify a consensus gene junction sequence in the genome fragments of VVeV1 (Table 2). Pairwise aa sequence identities between the VVeV1-encoded proteins and those from known dichorhaviruses showed low sequence identities of 20% or less in the P3 and G proteins and 20.5–23.1% in the N protein (Table 6). The phylogenetic analysis based on the N protein aa sequences showed that VVeV1 clusters with the dichorhaviruses but is placed by itself in a sister clade (Figure 2).

### 3.6. Varicose-Like Viruses

The complete coding sequences of five varicosa-like viruses, tentatively named *Allium angulosum* virus 1 (AAnV1), *Brassica rapa* virus 1 (BrRV1), *Lolium perenne* virus 1 (LoPV1), *Melampyrum roseum* virus 1 (MelRoV1) and *Pinus flexilis* virus 1 (PiFleV1), as well as the partial genomes of two others, named *Phlox pilosa* virus 1 (PhPiV1) and spinach virus 1 (SpV1), were assembled in this study (Table 1). The genomes of AAnV1, BrRV1, LoPV1, MelRoV1, PhPiV1 and SpV1 consist of two RNA segments. RNA 1 of AAnV1, BrRV1, LoPV1, MelRoV1 and SpV1 encode only the L protein, whereas RNA 1 of PhPiV1 likely carries an additional gene 3′ to the L gene (Figure 1C). RNA 2 of AAnV1, BrV1, LoPV1 and SpV1 has three genes in the order 3′-N-P2-P3-5′, RNA 2 of MelRoV1 has four genes in the order 3′-N-P2-P3-P4 -5′ and RNA 2 of PhPiV1 likely has five genes in the order 3′-N-P2-P3-P4-P5-5′ (Figure 1C). Based on our assemblies, the PiFleV1 genome does not appear to be segmented like the genomes of other varicosaviruses and has five ORFs in the order 3′-N-P2-P3-P4-L-5′ (Figure 1C), suggesting that varicosa-like viruses may also have unsegmented genomes.

The predicted LoPV1 N protein is the longest varicosavirus N protein described so far, whereas the MelRoV1 L protein is the smallest L protein described so far among the varicosaviruses, while the PiFleV1 L protein is the largest varicosavirus L protein described so far (Appendix A). Interestingly, the PiFLeV1 N protein is basic, whereas the N proteins of AAnV1, BrRV1, LoPV1, MelRoV1, PhPiV1 and SpV1 are acidic. Moreover, AAnV1 and BrR1 P2s are basic, while P2s of LoPV1, MelRoV1, PiFleV1 and SpV1 are acidic. Furthermore, P3s of AAnV1, MelRoV1, PhPiV1 and PiFleV1 are basic, whereas the BrRV1 and LoPV1 P3s are acidic (Appendix A).

Based on the NLS scores, AAnV1 L and P2, LoPV1 N and P2 and the MelRoV1 L protein are expected to be exclusively located in the nucleus. The BrRV1 N; BrRv1, PiFleV1 and SpV1 P2; PiFleV1 P3 and BrRv1, LoPV1 and SpV1 L proteins are predicted to localize to both the nucleus and the cytoplasm. Leucine-rich NESs are predicted in the BrRV1, MelRoV1, PhPiV1, PiFleV1 and SpV1 N proteins; the AAnV1, BrRV1, LoPV1, MelRoV1, PiFleV1 and SpV1 P2; the BrRV1, LoPV1, MelRoV1, PhPiV1, PiFleV1 and SpV1 P3 and the MelRoV1, PhPiV1 and PiFleV1 P4 (Appendix A). No transmembrane domain or signal peptide was detected in any of the proteins encoded by AAnV1, BrRV1, LoPV1, MelRoV1, PhPiV1, PiFleV1 and SpV1.

The consensus gene junction sequences 3′ AU(N)_5_UUUUUGCUCU 5′ of AAnV1, LoPV1, MelRoV1, PhP1V1 and SpV1 are identical and the same as in the genomes of the known varicosaviruses *Alopecurus myosuroides* varicosavirus 1 (AMVV1), lettuce big-vein associated virus (LBVaV, red clover-associated varicosavirus (RCaVV) and vitis varicosavirus (VVV). The gene junction sequences of BrRV1 and PiFleV1 differ slightly: the BrRV1 5′-end sequence is CUCA instead of CUCU, while the PiFleV1 3′-end sequence is GU(N)_5_UUUUU instead of AU(N)_5_UUUUU (Table 2).

The pairwise aa sequence identities between the cognate-encoded proteins of AAnV1, BrRV1, LoPV1, MelRoV1, PhPiV1, PiFleV1 and SpV1 and those from other varicosaviruses vary significantly (Table 7). The phylogenetic analysis based on the deduced N protein aa sequences placed all seven novel varicosa-like viruses into a clade with the known varicosaviruses. BrRV1 and SpV1 clustered with RCaVV, and these viruses have a similar genomic organization with the three ORFs on RNA 2. AAnVi clustered with the clade formed by LoPV1 and AMVV1, and these viruses infect monocots. PhPV1 clustered with LBVaV, while the unique PiFleV1 was placed in a sister clade to the clade composed by LBVaV and PhPV1, and MelRoV1 clustered with vitis varicosavirus (VVV) (Figure 2).

## 4. Discussion

In the last few years, several novel plant rhabdoviruses, which may be asymptomatic, have been reported in HTS studies [47,48,49,50,51,52,53,54,55]. Thus, it is tempting to speculate that some plant rhabdovirus-like sequences may be hidden in published plant transcriptome databases, generated with diverse objectives beyond virus research, where viral RNA was inadvertently copurified with endogenous plant RNA and sequenced. To prove this point, we and others have previously identified the sequences of a small number of plant rhabdoviruses contained in public transcriptome databases [8,9,16]; however, an extensive search has not been conducted to date. Therefore, we queried the publicly available plant transcriptome datasets in the transcriptome shotgun assembly (TSA) database hosted at NCBI, which resulted in the identification of 27 novel plant rhabdoviruses.

### 4.1. Discovery of Novel Plant Rhabdoviruses Expands the Diversity and Evolutionary History of Rhabdovirids

The coding-complete or complete genomic sequences of 66 plant rhabdoviruses were reported by early 2021. Thirty-three of them are cytorhabdoviruses, thirteen alphanucleorhabdoviruses, ten betanucleorhabdoviruses, five dichorhaviruses, four varicosaviruses and one gammanucleorhabdovirus. Similarly, half of the novel sequences reported in this study are those of putative cytorhabdoviruses, indicating the extensive diversification of this group of viruses. The novel 27 viruses discovered in this study all appear to be members of new species and account for nearly half of the plant rhabdovirus species reported so far. In addition, five new putative betanucleorhabdoviruses shed more light on the diversity and evolution of this recently created genus. One newly identified putative alphanucleorhabdovirus and one novel putative dichorhavirus expand the diversity of their respective genera. Finally, we detected seven novel varicosa-like viruses, which doubles the number of known varicosaviruses, providing new information about the genomic diversity and evolution of these little-studied fungi-transmitted viruses. Thus, our study provides the most complete insight to date about the genomic diversity and evolution of plant rhabdoviruses, complementing the status quo of these viruses with additional data on genomic organization and highlighting their apparent genetic flexibility. Overall, the sequence identity between the newly discovered viruses and those plant rhabdoviruses already described was low, a common feature among these taxa characterized by a high level of diversity in both genome sequence and organization [20]. The low sequence identity of the novel viruses with the closest previously described virus may indicate that there is still a significant amount of virus “dark matter” within the plant rhabdovirus space worth exploring that potentially contains a significant number of yet to be discovered plant rhabdoviruses. Future works should focus on the analysis of additional RNA datasets of diverse potential hosts of the partial virus genomes identified in this study.

Viruses assigned to different species within the genera *Alphanucleorhabdovirus*, *Betanucleorhabdovirus* and *Cytorhabdovirus* have a nucleotide sequence identity lower than 75% in the complete genome sequence and occupy different ecological niches as evidenced by differences in the host species and/or arthropod vectors (https://talk.ictvonline.org/ictv-reports/ictv_online_report/negative-sense-rna-viruses/w/rhabdoviridae) (accessed 11 April 2021). Cytorhabdoviruses may also have an aa sequence identity of less than 80% in all cognate ORFs. Viruses assigned to different species in the genus *Dichorhavirus* have less than 80% nucleotide sequence identity in the L gene. Based on these species demarcation criteria, ATV1 should be classified in a new *Alphanucleorhabdovirus* species and AscSyV2, PleArV1 and RhoDeV1 in a new species in the genus *Betanucleorhabdovirus*; due to incomplete sequences, PerMiV1 and CusReV1 cannot be classified based on these criteria. AntAmnV1, AscSyV1, BeTaV1, GlLV1, GymDenV1, LotCorV1, NymAV1, TaEV and TrAV1 should be classified in a new species in the genus *Cytorhabdovirus*, whereas AChV1, DiCoV1, PelRaV1 and SuSV1, due to incomplete genome sequence data, cannot be classified. The dichorha-like VVeV1 also awaits at least coding-complete sequence data before it can be formally classified in this genus. The species demarcation criteria recently proposed for the genus *Varicosavirus* by the ICTV *Rhabdoviridae* Study Group included a minimum nucleotide sequence divergence of 50% in the L protein. Based on this genetic distance, the varicosa-like viruses AAnV1, BrRV1, LoPV1, MelRoV1, PiFleV1 and SpV1 would not be considered different species, and PhPiV1 cannot be classified due to incomplete sequencing data. Considering the diverse host plant species these viruses were identified in and their phylogenetic relationships, these viruses should likely be classified in different species. Thus, we propose a nucleotide sequence identity of 75% across the genome and in the N gene as thresholds for species demarcation in the *Varicosavirus* genus, in line with the species demarcation criteria for the other genera of plant rhabdoviruses.

### 4.2. Host Range of the Novel Plant Rhabdoviruses

Most of the plant hosts in which the novel viruses of this study were identified are dicots; nevertheless, AChV1, which was detected in Chinese onions, is likely the first monocot-infecting cytorhabdovirus that belongs to a clade of aphid-transmitted viruses. Furthermore, most of the source hosts in which the novel viruses were identified are herbaceous plants, which, overall, are the most common hosts of plant rhabdoviruses [18]. However, one novel putative varicosavirus, *Pinus flexilis* virus 1, was associated with a gymnosperm host. Few viruses were identified previously from gymnosperms [12,56,57,58,59]. Interestingly, a putative RdRp sequence was identified in the gymnosperm *Sciadopitys verticillata* and proposed to belong to a varicosavirus [14]. Therefore, to our knowledge, this study is the first to report the complete coding sequence of a plant rhabdovirus associated with a gymnosperm host and presenting a unique unsegmented genome, redefining the genome architecture of varicosaviruses. Future virus discovery studies should focus on gymnosperm viromes, which have been understudied and likely hide a rich and diverse number of novel viruses.

### 4.3. Diverse Genome Organizations of Plant Rhabdoviruses

Interestingly, the genome sequence of four novel viruses discovered in this study, GymDenV1 (only four predicted genes), TaEV and TrAV1 (both with only five predicted genes) and PiFleV1 (an unsegmented varicosa-like virus), have a unique genome architecture among plant rhabdoviruses that differs from the consensus genome organization reported for previously known cytorhabdoviruses and varicosaviruses [21]. The genomes of the cytorhabdoviruses GymDenV1, TaEV1 and TrAV1 lack a glycoprotein ORF, which, for rhabdoviruses, is not essential for replication and systemic movement, as demonstrated using an infectious clone of Sonchus yellow net virus [60]. Furthermore, isolates of citrus-associated rhabdovirus (CiaRV) were recently shown to have an impaired G ORF [61].

Five canonical structural protein genes 3′-N-P-M-G-L-5′ are thought to be conserved among all rhabdoviruses [20], while plant rhabdovirus genomes contain at least six ORFs [21]. However, only four genes were predicted in the genome of the varicosavirus RCaVV [51]. Four genes were also identified in some of the varicosa-like virus genomes assembled in this study, suggesting that a minimal set of four genes may be sufficient for varicosaviruses to replicate in a plant host. This hypothetical minimal requirement appears to also apply to cytorhabdoviruses. For example, GymDenV1 encodes only the nucleocapsid core (NC) proteins N, P and L that are essential for virus replication and transcription and the M protein that is required for condensation of the core during virion assembly [20]. Thus, how GymDenV1 moves from cell to cell remains to be unraveled. Interestingly, no cell-to-cell movement protein has been identified so far in the varicosaviruses.

The discovery of diverse novel rhabdo-like viruses in metagenomics studies of arthropods [62] supports the assumption that arthropods have been fundamental to rhabdovirus evolution [63], and the G protein was found to be essential for virus attachment to predict cellular receptors in the midgut of arthropod vectors that facilitate virus uptake [20]. This agrees with the hypothesis that cytorhabdoviruses, nucleorhabdoviruses and dichorhaviruses evolved from viruses of plant-feeding arthropods that acquired movement proteins and assorted RNAi suppressors through recombination with preexisting plant viruses [3]. The viruses in these three genera appear to be the least plant-specialized among the negative-sense RNA viruses. However, the evolution of the nonenveloped negative-sense RNA plant viruses, such as the fungal-transmitted varicosaviruses, which likely do not encode a G protein, clearly reflects the adaptation to a plant-specific lifestyle, raising the possibility that their origin is via a *trans*-kingdom horizontal transfer between fungi and plants [3]. A G protein-defective genome was recently identified in citrus isolates of CiaRV, and the authors speculated that the eventual “simplification” of viral genomes to adapt to plants without requiring an arthropod vector could provide an evolutionary advantage, especially in fruit trees that are propagated artificially by asexual modes, such as cutting and grafting [61]. Nevertheless, the tentative hosts of GymDenV1, TaEV1 and TrAV1 are herbaceous plants; thus, an evolutionary advantage linked to the lack of a G protein is not obvious for these viruses. Strikingly, TaEV1 is phylogenetically related to arthropod-transmitted cytorhabdoviruses. Thus, further studies should focus on the potential vector and the mode of transmission of GymDenV1, TaEV1 and TrAV1 to complement their peculiar minimalist genome organization and evolutionary links with biological data.

### 4.4. Phylogenetic Relationships among Plant Rhabdoviruses as Predictors for Vector Types

Among all plant rhabdoviruses studied so far, there is a strong correlation between phylogenetic relationships and vector types [18,20]. We therefore predict that the novel betanucleorhabdoviruses are likely aphid-transmitted, while the putative dichorhavirus VVeV1 is likely transmitted by *Brevipalpus* mites, and the varicosa-like viruses may be transmitted by chytrid fungi. Based on its phylogenetic clustering, the alphanucleorhabdovirus ATV1 is likely transmitted by a planthopper. Among the novel cytorhabdoviruses identified in this study, AChV1, AscSyV1, GlLV1, LotCorV1, NymAV1, PelRaV1 and SuSV1 are likely aphid-transmitted, while the vectors for AntAmV1, DiCoV1, GymDenV1, TaEV1 and TrAV1 cannot be predicted. The BeTaV1 genome was assembled from whitefly transcriptome data and, therefore, is likely a whitefly-transmitted cytorhabdovirus. The L protein-like TSA sequences included in the assembly of the BeTaV1 genome were previously reported to be 95–98% identical to soybean blotchy mosaic virus partial L gene sequences [18]; thus, it is tempting to speculate that the plant host of BeTaV1 may be soybeans. Recently, it was reported that the bean-associated cytorhabdovirus is efficiently transmitted by whiteflies, which was the first report of a whitefly-transmitted rhabdovirus [64], thus supporting our hypothesis that soybean blotchy mosaic virus may represent the second whitefly-transmitted rhabdovirus.

At least one transmembrane domain was identified in each P′ protein predicted in the genomes of a group of cytorhabdoviruses discovered in this study, consistent with a previous analysis of other cytorhabdoviruses, which also identified at least one transmembrane domain in every P′ protein [65]. Interestingly, this overprinted accessory protein is encoded by every cytorhabdovirus that appears to be aphid-transmitted. On the other hand, the P′ protein is not encoded in the genomes of aphid-transmitted betanucleorhabdoviruses. Therefore, the potential function of the P′ protein is unlikely to be directly associated with the vector specificity.

### 4.5. Cytorhabdoviruses

Two of the cytorhabdoviruses, AChV1 and SuSV1, whose genomes were only partially assembled, clustered phylogenetically in a monophyletic clade with ADV, ChYDaV, RVCV and SCV, all the viruses that contain a P6 accessory ORF between the G and L genes [66,67,68,69]. Given this clustering, it is likely that AChV1 and SuSV1 will have a similar genomic organization. At least one transmembrane domain has been predicted in every cytorhabdovirus with an accessory P6 ORF. Transmembrane domains were also predicted in the accessory ORF between the G and L genes of other cytorhabdoviruses [49,65,70]. Thus, the protein encoded by this small ORF may have membrane-associated functions similar to viroporins in vertebrate rhabdoviruses [71].

The phylogenetic relationships of the now expanded number of known cytorhabdoviruses provide some support for splitting the genus *Cytorhabdovirus* to establish three genera that represent different evolutionary lineages: (I) *Alphacytohabdovirus* would include species for all aphid-transmitted cytorhabdoviruses; (II) *Betacytorhabdovirus* would include species for all those cytorhabdoviruses likely transmitted by leafhoppers, planthoppers and whiteflies, as well as AntAmnV1, DiCoV1 and TaEV1, and (III) *Gammacytorhabdovirus* would include species for GymDenV1 and TrAV1 (Figure 2).

### 4.6. Nucleorhabdoviruses

The ATV1 genome that was assembled from the transcriptome data of the monocot agave does not encode any accessory ORFs, like most of the alphanucleorhabdoviruses, except the group of PYDV-like viruses that encode an X ORF of unknown function [18]. ATV1, while branching into a sister clade, appears to have a close evolutionary relationship with the cluster of planthopper-transmitted monocot-infecting alphanucleorhabdoviruses MIMV, MMV, MMaV and TaVCV. It is tempting to speculate that this clade evolved from a common ancestor that adapted to infect monocots and to be transmitted by planthoppers. Furthermore, the isoelectric point (IEP) of ATV1-predicted proteins is similar to that of MIMV, MMV, MMaV and TaVCV-predicted proteins, thus supporting a link between these viruses and ATV1.

All the betanucleorhabdoviruses identified in this study, AscSyV2, CusReV1, PerMiV1, PleArV1 and RhoDeV1, appear to be associated with dicot hosts, which is consistent with every betanucleorhabdovirus reported so far [18]. The dicot host range of the betanucleorhabdoviruses is likely linked to their insect vector, given that every betanucleorhabdovirus where a vector has been experimentally determined is transmitted by aphids [18]. Although aphids can colonize both dicots and monocots [72], it has been suggested that these insects more successfully feed on dicots [73]. Regarding their genome organization, the five novel betanucleorhabdoviruses identified in this study encode six plant rhabdovirus ORFs in the conserved order 3′-N-P-P3-M-G-L-5′ without any accessory ORFs, like the majority of previously known betanucleorhabdoviruses, such as the well-studied SYNV. CusReV1, PerMiV1, PleArV1 and RhoDeV1 likely belong to the same evolutionary lineage as BFTaV1, BCaRV, CdVCV1, DYVV, GSPNuV, SYNV and SYVV, which may represent the ancestral clade within the betanucleorhabdoviruses. However, AscSyV2′s closest evolutionary relationship is with the betanucleorhabdoviruses ApRVA and AaNV, which differ in their genome organization by having an accessory ORF located between the M and G genes [46,74]. Interestingly, the IEP of the AscSyV2, CusReV1, PerMiV1, PleArV1 and RhoDeV1 N proteins is different to the IEP of the cognate proteins encoded by the phylogenetically related betanucleorhabdoviruses. It is unknown if this difference in the IEP may have a biological effect in terms of RNA or protein binding. Moreover, the RhoDeV1 M protein is acidic, while the BFTaV1, BCaRV, BmV2, CdVCV1, CusReV1, DYVV, GSPNuV, PleArV1, SYNV and SYVV N proteins are basic. It has been suggested that charge differences in the M proteins may be associated with differences in their abilities to interact with the negatively charged lipids of the membrane [75]. NLS and NES were predicted for most of the AscSyV2, ATV1, CusReV1, PerMiV1, PleArV1 and RhoDeV1-encoded proteins, which can be expected, since nucleorhabdoviruses replicate in the nuclei of infected cells [20].

The phylogenetic relationships of nucleorhabdoviruses, including the novel alpha- and betanucleorhabdoviruses discovered in our analysis, clearly support the recent split of the previous genus *Nucleorhabdovirus* into the three genera *Alphanucleorhabdovirus*, *Betanucleorhabdovirus* and *Gammanucleorhabdovirus* [18].

### 4.7. Varicosaviruses

Currently, there are only three varicosaviruses recognized by the ICTV, LBVaV, AMVV1 and RCaVV, and a novel varicosavirus, VVV, was recently identified [52]. Nevertheless, the available information regarding varicosavirus gene functions is generally scarce, and the functional roles of the varicosavirus P2, P3, P4, P5 and P6 proteins remain to be elucidated. No transmembrane domain was predicted in any of the proteins encoded by the varicosavirus genomes identified in this study, as well as in those already described; thus, it appears that no varicosavirus-encoded protein has a membrane-associated function. The AAnV1, L and P2 proteins; LoPV1 N and P2 proteins and AMVV1 and MelRoV1 L proteins are predicted to be exclusively located in the nucleus, whereas NES are predicted in most N, P2 and P3 proteins and some P4 proteins (Appendix A). This suggests a potential role of these proteins in the cell nucleus.

The N protein encoded by the unsegmented PiFLeV1 is basic; on the other hand, the N proteins encoded by the bisegmented varicosaviruses AAnV1, AMVV1, BrRV1, LBVaV, LoPV1, MelRoV1, PhPiV1, RCaVV, SpV1 and VVV are acidic. This difference in the IEP may be associated with a different replication mechanism of the unsegmented PiFLeV1 compared to the segmented varicosaviruses. Moreover, AAnV1 and BrRV1 P2s are basic, while the P2s of AMVV1, LBVaV, LoPV1, MelRoV1, PiFleV1, RCaVV, SpV1 and VVV are neutral or acidic. The biological significance of this difference is unknown, because the functional role of P2 still needs to be unraveled.

LoPV1 and AMVV1 are phylogenetically closely related; their N and P3 aa sequences are >50% identical, and both are associated with grasses in the family *Poaceae*. However, LoPV1 RNA1 has one ORF, while AMVV1 RNA1 has two [76]. These monocot-infecting varicosaviruses are phylogenetically related to AAnV1, which also is associated with a monocot host and shares a similar genomic organization with LoPV1. PhPiV1 clustered with LBVaV; thus, it is likely that its genome, which was partially assembled, has a similar genomic organization to LBVaV, with two ORFs encoded in RNA1 and five in RNA2 [77]. Interestingly, the RNA2 of four of the novel varicosa-like viruses identified in this study have three ORFs, similar to RCaVV and AMVV1 RNA2 [51,76]. On the other hand, the RNA2 of one novel varicosa-like virus identified in our study has four ORFs. Thus, the number of ORFs identified in the RNA2 of every novel varicosa-like virus reported in this study, except for PhPiV1, is different from the first identified varicosavirus, LBVaV, and the recently described varicosavirus, VVV, which have five ORFs [52,77]. The RNA1 segments of AAnV1, BrRV1, LoPV1, SpV1 and MelRoV1 are similar to the RNA1 of RCaVV and VVV in that they only encode the L protein, whereas a small ORF before the L gene was identified in PhPiV1 and previously in AMVV1 and LBVaV [51,52].

Among the varicosa-like viruses identified in this study, PiFleV1 is unique in terms of genome organization, since its genome is unsegmented, a characteristic that differs from the currently known bisegmented genome architecture of varicosaviruses [21]. This may be an adaptation to its gymnosperm host. Unfortunately, we were not able to assemble the complete genome of the varicosavirus associated with the gymnosperm *Sciadopitys verticillata* previously reported by Mushegian and colleagues [14] to support this hypothesis. Based on genome organization and phylogenetic placement, PiFleV1 appears to be the first example where rhabdoviruses with segmented and unsegmented genomes are closely related and may be classified in the same genus.

## 5. Conclusions

In summary, this study illustrates the complexity and diversity of plant rhabdoviruses genomes and demonstrates that analyzing SRA public data is a valuable tool not only to accelerate the discovery of novel viruses but, also, to gain insight into their evolution and to refine virus taxonomy. However, the inability to go back to the biological material to confirm viral genome sequences and to link the presence of the viruses to a specific host is the main drawback of the data mining approach for virus discovery. This limitation could lead to the potential misidentification of host species linked to viruses. Thus, researchers need to be cautious when analyzing SRA public data for virus discoveries.

## Figures and Tables

**Figure 1 viruses-13-01304-f001:**
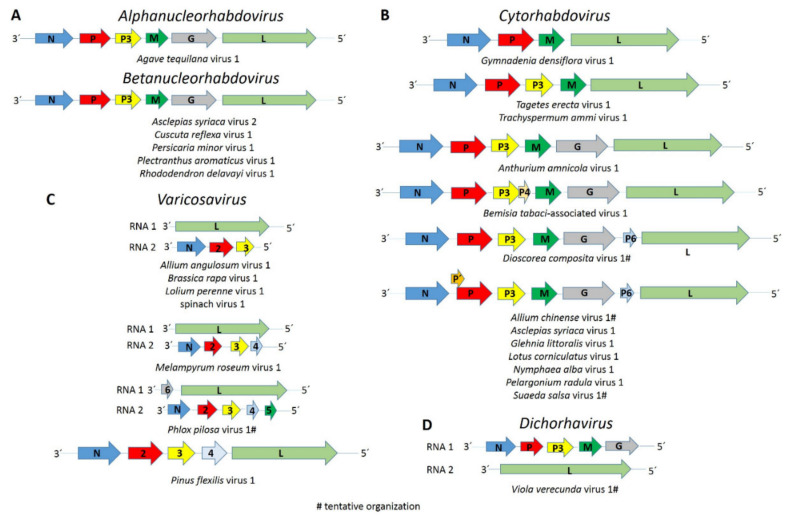
Diagrams depicting the genomic organization of each (**A**) alpha- and betanucleorhabdovirus, (**B**) cytorhabdovirus, (**C**) varicosavirus and (**D**) dichorhavirus sequence assembled in this study. Abbreviations: N, nucleoprotein coding sequence (CDS); P, phosphoprotein CDS; P3, putative cell-to-cell movement protein CDS; M, matrix protein CDS; G, glycoprotein CDS; L, RNA-dependent RNA polymerase CDS.

**Figure 2 viruses-13-01304-f002:**
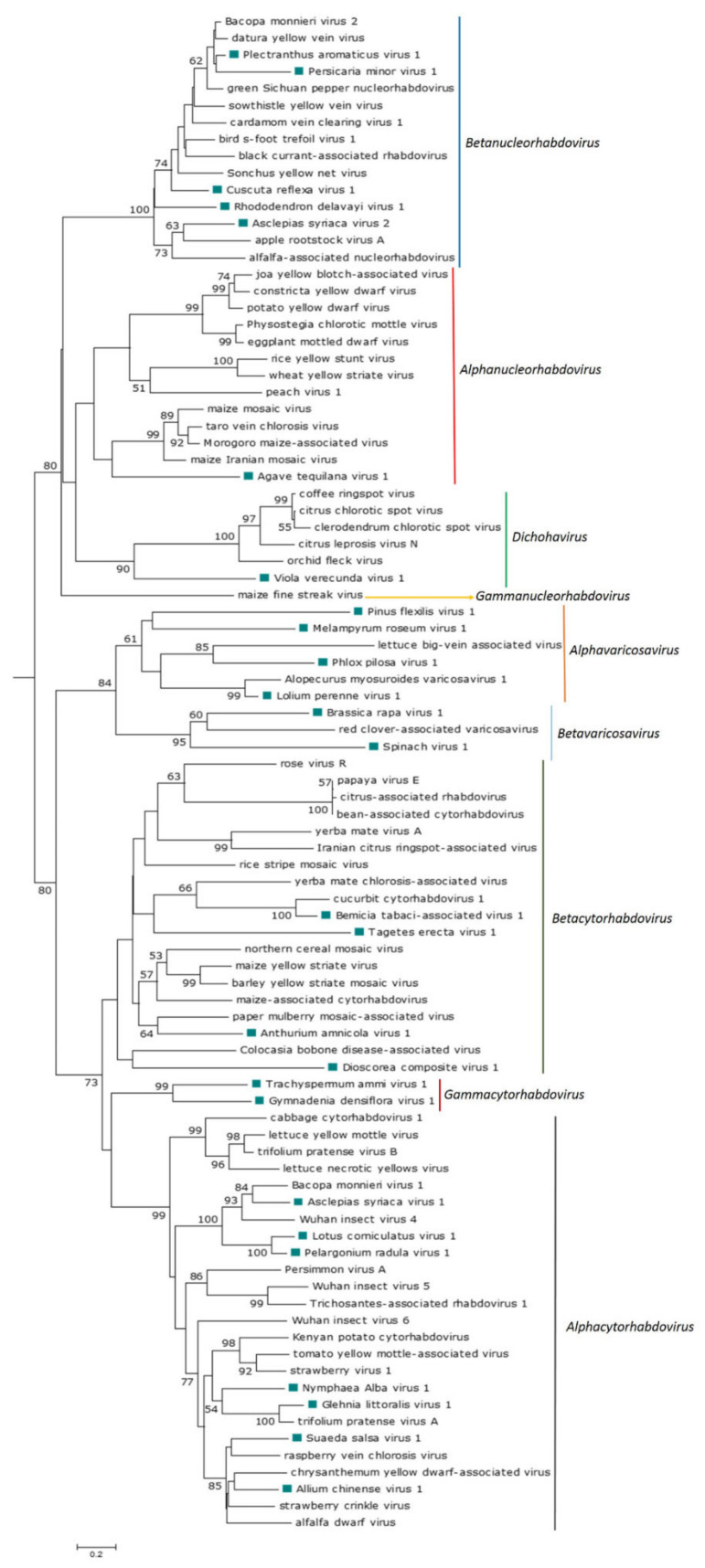
Maximum Likelihood phylogenetic tree based on a multiple amino acid alignment of the N protein sequences of plant rhabdoviruses constructed with the WAG + G + F model. Bootstrap values following 1000 replicates are given at the nodes, but only the values above 40% are shown. The scale bar indicates the number of substitutions per site. The viruses identified in this study are noted with green rectangles. The accession numbers of every virus used to construct the ML tree are listed in Appendix A.

**Table 2 viruses-13-01304-t002:** Consensus plant rhabdovirus gene junction sequences.

Genus	Virus *	3′End mRNA	Intergenic Spacer	5′End mRNA
*Alphanucleorhabdovirus*	ATV1	AUUCUUUUU	GGG	UUG
	CYDV	AUUAUUUUU	GGG	UUG
	EMDV	AUUAUUUUU	GGG	UUG
	JYBaV	AUUAUUUUU	GGG	UUG
	MIMV	AUUCUUUUU	GGG	UUG
	MMV	AUUCUUUUU	GGG	UUG
	MMaV	AUUCUUUUU	GGG	UUG
	PeV1	AUUU(A/C)UUUU	G(N)n	UUG
	PhCMoV	AUUAUUUUU	GGG	UUG
	PYDV	AUUAUUUUU	GGG	UUG
	RYSV	AUUAUUUUU	GGG	UUG
	TaVCV	AUUCUUUUU	GGG	UUG
	WYSV	UAAAUUUUU	GGGG	UUG
*Betanucleorhabdovirus*	AscSyV2	AUUCUUUUU	GG	UUG
	CusReV1	AUUCUUUUU	GG	UUG
	PerMiV1	AUUCUUUUU	GG	UUG
	PleArV1	AUUCUUUUU	GG	UUG
	RhoDeV1	AUUCUUUUU	GG	UUG
	AaNV	AUUCUUUUU	GG	UUG
	ApRVA	AUUCUUUUU	GG	UUG
	BFTaV	AUUCUUUUU	GG	UUG
	BCaRV	AUUCUUUUU	GG	UUG
	BmV2	AUUCUUUUU	GG	UUG
	CdVCV1	AUUCUUUUU	GG	UUG
	DYVV	AUUCUUUUU	GG	UUG
	GSPNuV	AUUCUUUUU	GG	UUG
	SYNV	AUUCUUUUU	GG	UUG
	SYVV	AUUCUUUUU	GG	UUG
*Gammanucleorhabdovirus*	MFSV	AUUUAUUUU	GUAG	UUG
*Dichorhavirus*	VVeV1	-	-	-
	CiCSV	AUUUAUUUU	GUAG	UU
	CiLV-N	AUUUAUUUU	GUAG	UU
	ClCSV	AUUUAUUUU	GUAG	UU
	CoRSV	AUUUAUUUU	GUAG	UU
	OFV	AUUUAUUUU	GUUG	UU
*Cytorhabdovirus*	AChV1	AAUUAUUUU	GAU	CUA
	AntAmV1	AUUAUUUUU	GCU	CUU
	AscSyV1	AAUUAUUUU	GNU	CNN
	BeTaV1	UUAUUUUUU	GA	CUC
	DiCoV1	AUAUUUUUU	GG(N)_n_	CNN
	GlLV1	AAUUAUUUU	GAU	CUU
	GymDenV1	AAUCUUUUU	A(N)_n_	CNN
	LotCorV1	AAUUAUUUU	GGU(N)_n_	CUG
	NymAV1	AUUAAUUUU	GAU	CUN
	PelRaV1	AAUUAUUUU	GGU(N)_n_	CUG
	SuSV1	AAUUAUUUU	GAU	CUU
	TaEV1	AUUCUUUUU	GG(N)_n_	CUN
	TrAV1	AUUCUUUUU	A(N)_n_	CNU
	AcCV1	AAUUAUUUU	GAU	CUG
	ADV	AAUUAUUUU	GAU	CUU
	BmV1	AAUUAUUUU	GAN	CUG
	BYSMV	AUUAUUUUU	GA	CUC
	CCyV1	AAUUCUUUU	G(N)_n_	CUU
	CBDaV	AUUCUUUUU	GG	CUC
	ChYDaV	AAUUAUUUU	GAU	CUN
	CuCV1	AUUAUUUUU	GA	CUC
	KePCyV	AAUUAUUUU	GAU	CUU
	LNYV	AAUUCUUUU	G(N)_n_	CUU
	LYMoV	AAUUCUUUU	G(N)_n_	CUN
	MaCyV	AUUCUUUUU	GA	CUC
	MYSV	AUUAUUUUU	GA	CUC
	NCMV	AUUCUUUUU	GA	CUC
	PMuMaV	AUUAUUUUU	G(N)_n_	CUA
	PpVE	AUUCUUUUU	GAC	CCU
	PeVA	AAUUAUUUU	G(N)_n_	CUN
	RVR	AUUUAUUUU	GACU	CUA
	RSMV	AUUCUUUUU	GCU	CUG
	RVCV	AUUUAUUUU	GAU	CUU
	SCV	AAUUAUUUU	GAU	CUU
	StrV1	AAUUAUUUU	GAU	CUU
	TpVA	AAUUAUUUU	GAU	CUU
	TpVB	AAUUCUUUU	G(N)_n_	CUN
	TrARV1	AAUUAUUUU	GAU	CUU
	TYMaV	AAUUAUUUU	GAU	CUU
	WhIV4	AAUUAUUUU	GNU	CUU
	WhIV5	AAUUAUUUU	GAU	CNN
	WhIV6	AAUUAUUUU	GAU	CUN
	YmCaV	UUAUUUUUU	GA	CUC
	YmVA	AUUCUUUUU	GGU	CCU
*Varicosavirus*	AAnV1	AU(N)_5_UUUUU	G	CUCU
	BrRV1	AU(N)_5_UUUUU	G	CUCA
	LoPV1	AU(N)_5_UUUUU	G	CUCU
	MelRoV1	AU(N)_5_UUUUU	G	CUCU
	PhPiV1	AU(N)_5_UUUUU	G	CUCU
	PiFleV1	GU(N)_5_UUUUU	G	CUCU
	SpV1	AU(N)_5_UUUUU	G	CUCU
	AMVV1	AU(N)_5_UUUUU	G	CUCU
	LBVaV	AU(N)_5_UUUUU	G	CUCU
	RCaVV	AU(N)_5_UUUUU	G	CUCU
	VVV	AU(N)_5_UUUUU	G	CUCU

The consensus gene junction sequences of the viruses identified in this study are highlighted in light grey. * Names and abbreviations of newly identified viruses are listed in Table 1, while the names and abbreviations of known viruses are listed in Appendix A.

**Table 3 viruses-13-01304-t003:** Pairwise identity percentages between ATV1 and the selected alphanucleorhabdoviruses.

		Genome ^a^	N ^b^	P ^b^	P3 ^b^	M ^b^	G ^b^	L ^b^
ATV1 vs.	MMaV	47.0	32.4	16.7	17.5	17.5	29.3	35.0
	MMV	49.3	29.1	13.7	20.5	16.4	29.1	34.5
	PeV1	49.1	26.9	18.8	20.1	14.3	28.7	32.9
	PYDV	48.8	24.1	10.9	18.2	13.6	23.5	31.8

^a^ Nucleotide percentages; ^b^ amino acid percentages; N/C: not complete; virus names are listed in Appendix A and Table 1.

**Table 4 viruses-13-01304-t004:** Pairwise identity percentages between AscSyV2, CusReV1, PleArV1 and RhoDeV1 and the selected betanucleorhabdoviruses.

		Genome ^a^	N ^b^	P ^b^	P3 ^b^	M ^b^	G ^b^	L ^b^
AscSyV2 vs.	ApRVA	51.3	47.5	24.3	26.3	23.6	33.2	44.2
	BCaRV	50.4	40.5	23.4	25.0	22.8	30.3	38.5
	CusReV1	N/C	40.9	23.0	18.1	N/C	N/C	N/C
	DYVV	50.6	42.3	19.5	22.2	20.5	27.0	38.1
	GSPNuV	46.3	42.7	21.7	19.9	20.1	29.0	38.7
	PleArV1	49.3	41.1	19.6	19.3	19.1	26.4	37.2
	RhoDeV1	51.4	40.7	18.1	25.5	19.3	27.1	39.9
CusReV1 vs.	ApRVA	N/C	42.5	20.8	19.6	N/C	N/C	N/C
	AscSyV2	N/C	40.9	23.0	18.1	N/C	N/C	N/C
	BCaRV	N/C	48.3	27.6	28.4	N/C	N/C	N/C
	DYVV	N/C	59.0	32.6	41.7	N/C	N/C	N/C
	GSPNuV	N/C	59.0	30.8	44.0	N/C	N/C	N/C
	PleArV1	N/C	60.5	31.6	40.8	N/C	N/C	N/C
	RhoDeV1	N/C	38.6	24.8	23.0	N/C	N/C	N/C
PleArV1 vs.	ApRVA	49.8	44.0	22.2	20.6	16.7	27.2	37.4
	AscSyV2	49.3	41.1	19.6	19.3	19.1	26.4	37.2
	BCaRV	52.1	49.9	30.1	26.5	31.6	36.4	47.3
	CusReV1	N/C	60.5	31.6	40.8	N/C	N/C	N/C
	DYVV	62.1	70.7	45.6	63.2	48.1	61.0	62.2
	GSPNuV	45.8	64.5	34.8	43.7	41.0	48.4	55.4
	RhoDeV1	50.9	42.5	22.7	29.8	25.4	31.9	42.8
RhoDeV1 vs.	ApRVA	51.7	40.4	23.2	22.4	22.2	30.2	40.8
	AscSyV2	51.4	40.7	18.1	25.5	19.3	27.1	39.9
	BCaRV	53.1	42.0	27.2	34.5	27.4	33.6	45.0
	CusReV1	N/C	38.6	24.8	23.0	N/C	N/C	N/C
	DYVV	51.3	42.2	23.2	27.5	26.5	34.4	43.7
	GSPNuV	46.3	41.1	22.4	25.4	28.6	32.8	43.3
	PleArV1	50.9	42.5	22.7	29.8	25.4	31.9	42.8

^a^ Nucleotide percentages; ^b^ amino acid percentages; N/C: not complete; virus names are listed in Appendix A and Table 1.

**Table 5 viruses-13-01304-t005:** Pairwise identity percentages between the cytorhabdoviruses assembled in this study and the selected cytorhabdovirus.

		Genome ^a^	N ^b^	P ^b^	P3 ^b^	P4 ^b^	M ^b^	G ^b^	P6 ^b^	L ^b^
AChV1 vs.	ADV	N/C	44.3	30.8	34.5		27.9	N/C	N/C	N/C
	ChYDaV	N/C	41.5	38.3	44.9		32.4	N/C	N/C	N/C
	RVCV	N/C	42.2	29.6	35.1		22.8	N/C	N/C	N/C
	SCV	N/C	48.4	29.9	38.2		26.9	N/C	N/C	N/C
	SuSV1	N/C	N/C	N/C	42.7		36.7	N/C	N/C	N/C
AntAmV1 vs.	CBDaV	51.2	31.7	21.7	21.0		18.8	21.5	-	39.7
	MaCyV	50.8	28.3	19.3	17.9		16.4	22.1	-	40.2
	PMuMaV	51.8	41.0	25.6	29.3		35.3	36.4	-	45.5
AscSyV1 vs.	BmV1	54.8	50.4	37.6	55.3		34.5	44.3	27.6	53.9
	PelRaV1	N/C	39.7	31.9	35.6		22.0	28.9	20.2	N/C
	WhIV4	53.2	48.7	35.3	54.2		30.8	41.8	22.8	50.1
BeTaV1 vs.	CuCVi1	61.8	68.2	64.8	53.5	45.6	61.2	60.4	-	72.5
	DiCoV1	N/C	19.4	N/C	N/C	-	14.9	17.2	-	N/C
	YmCaV	51.5	30.6	20.3	31.7	19.9	21.2	30.8	-	42.4
DiCoV1 vs.	MYSV	N/C	23.3	N/C	N/C	-	20.2	21.0	17.3	N/C
	CBDaV	N/C	25.2	N/C	N/C	-	23.8	22.5	-	N/C
	RVR	N/C	20.7	N/C	N/C	-	17.6	18.7	19.0	N/C
GlLV1 vs.	NymAV1	49.7	40.0	23.8	27.3	-	24.0	37.6	24.1	50.0
	StrV1	49.9	38.7	25.0	37.0	-	26.5	42.1	22.4	50.5
	TpVA	65.6	67.8	60.9	78.1	-	66.7	71.6	75.8	76.1
	TYMaV	45.9	36.2	21.6	34.2	-	20.7	44.1	11.9	49.1
GymDenV1 vs.	LYMoV	43.5	22.0	19.0	-	-	19.4	-	-	29.5
	SCV	40.0	23.5	17.2	-	-	18.7	-	-	30.1
	TrAV1	53.8	34.4	23.1	-	-	26.8	-	-	56.3
	WhIV5	44.4	19.9	15.9	-	-	10.7	-	-	30.6
LotCorV1 vs.	AscSyV1	50.3	39.9	28.5	40.5	-	20.2	30.1	18.5	43.5
	PelRaV1	N/C	61.8	44.7	58.6	-	55.9	47.9	31.6	N/C
	WhIV4	52.0	40.9	29.7	39.7	-	20.8	30.7	12.5	43.8
NymAV1 vs.	GlLV1	49.7	40.0	23.8	27.3	-	24.0	37.6	24.1	50.0
	KePCyV	52.9	39.3	23.6	32.9	-	23.7	40.7	19.4	47.9
	StrV1	51.2	38.6	22.6	24.5	-	19.8	40.5	18.0	48.0
	TpVA	53.8	39.2	24.9	30.8	-	23.6	37.7	17.0	50.1
PelRaV1 vs.	AscSyV1	N/C	39.7	31.9	35.6	-	22.0	28.9	20.2	N/C
	LotCorV1	N/C	61.8	44.7	58.6	-	55.9	47.9	31.6	N/C
	WhIV4	N/C	40.5	28.5	39.6	-	20.4	27.5	11.7	N/C
SuSV1 vs.	AChV1	N/C	N/C	N/C	42.7	-	36.7	N/C	N/C	N/C
	ADV	N/C	N/C	N/C	37.0	-	30.2	N/C	N/C	N/C
	ChYDaV	N/C	N/C	N/C	37.9	-	35.6	N/C	N/C	N/C
	RVCV	N/C	N/C	N/C	40.2	-	30.8	N/C	N/C	N/C
	SCV	N/C	N/C	N/C	36.8	-	26.5	N/C	N/C	N/C
TaEV1 vs.	MYSV	50.3	23.3	15.2	16.2	-	15.0	-	-	41.7
	PpVE	47.4	21.3	17.6	15.1	-	14.1	-	-	34.9
	RSMV	48.7	25.0	15.7	17.8	-	16.3	-	-	38.4
	RVR	48.5	23.8	16.8	26.7	-	21.0	-	-	38.6
TrAV1 vs.	CCyV1	46.3	23.0	16.0	18.7	-	16.3	-	-	30.2
	GymDenV1	53.8	34.4	23.1	-	-	26.8	-	-	56.3
	StrV1	44.7	24.6	16.7	16.1	-	15.9	-	-	30.0
	WhIV5	46.9	20.5	19.3	11.2	-	13.3	-	-	29.6

^a^ Nucleotide percentages; ^b^ amino acid percentages; N/C: not complete; virus names are listed in Appendix A and Table 1.

**Table 6 viruses-13-01304-t006:** Pairwise identity percentages between VVeV1 and the dichorhaviruses.

		Genome ^a^	N ^b^	P ^b^	P3 ^b^	M ^b^	G ^b^	L ^b^
VVeV1 vs.	CiCSV	N/C	21.3	N/C	18.5	N/C	19.0	N/C
	CiLV-N	N/C	20.8	N/C	18.8	N/C	19.6	N/C
	ClCSV	N/C	21.1	N/C	18.2	N/C	19.2	N/C
	CoRSV	N/C	20.5	N/C	18.4	N/C	18.7	N/C
	OFV	N/C	23.1	N/C	19.1	N/C	20.1	N/C

^a^ Nucleotide percentages; ^b^ amino acid percentages; N/C: not complete; virus names are listed in Appendix A and Table 1.

**Table 7 viruses-13-01304-t007:** Pairwise identity percentages between the varicosaviruses assembled in this study and the reported varicosaviruses.

		N ^b^	2 ^b^	3 ^b^	4 ^b^	L ^a/b^
AAV1 vs.	AMVV1	25.9	20,2	20.2	-	54.4/37.8
	BrRV1	20.5	19.4	18.3	-	58.2/40.7
	LBVaV	20.9	19.7	12.6	-	54.1/37.7
	LoPV1	23.2	19.9	19.1	-	57.3/39.4
	MelRoV1	20.3	17.4	8.4	-	57.6/39.5
	PhPiV1	18.3	N/C	7.4	-	N/C
	PiFleV1	17.3	17.1	7.2	-	52.2/32.8
	RCaVV	21.4	19.8	18.7	-	56.2/38.9
	SpV1	19.3	18.5	17.4	-	57.1/39.3
BrRV1 vs.	AAnV1	20.5	19.4	18.3	-	58.2/40.7
	AMVV1	22.5	10.9	18.3	-	52.8/36.9
	LBVaV	19.9	15.4	6.7	-	53.6/37.3
	LoPV1	21.5	21.0	19.0	-	55.2/37.5
	MelRoV1	23.0	16.4	7.8	-	55.8/42.0
	PhPiV1	18.8	N/C	6.9	-	N/C
	PiFleV1	22.9	17.5	6.6	-	52.3/32.2
	RCaVV	26.9	17.7	21.8	-	59.6/50.5
	SpV1	24.6	18.1	N/C	-	59.8/51.5
LoPV1 vs.	AAnV1	23.2	19.9	19.1	-	57.3/39.4
	AMVV1	52.0	32.7	50.9	-	67.6/71.5
	BrRV1	21.5	21.0	19.0	-	55.2/37.5
	LBVaV	19.1	15.9	11.3	-	54.8/37.0
	MelRoV1	19.4	16.6	5.9	-	55.1/37.8
	PhPiV1	21.2	N/C	6.3	-	N/C
	PiFleV1	17.1	17.1	8.8	-	52.5/32.1
	RCaVV	18.8	9.6	18.5	-	54.5/37.5
	SpV1	18.8	16.8	N/C	-	54.1/35.9
MelRoV1 vs.	AAnV1	20.3	17.4	8.4	-	57.6/39.5
	AMVV1	22.6	16.8	6.7	-	54.8/38.0
	BrRV1	23.0	16.4	7.9	-	55.8/42.0
	LBVaV	19.6	16.0	24.0	11.1	53.6/37.7
	LoPV1	19.4	16.6	5.9	-	55.1/37.8
	PhPiV1	20.8	N/C	29.1	N/C	N/C
	PiFleV1	19.7	17.4	27.7	14.2	50.9/31.5
	RCaVV	22.5	15.3	9.9	-	55.2/41.0
	SpV1	21.9	20.5	N/C	-	56.3/41.0
PhPiV1 vs.	AAnV1	18.3	N/C	7.4	-	N/C
	AMVV1	22.5	N/C	7.1	-	N/C
	BrRV1	18.8	N/C	6.9	-	N/C
	LBVaV	30.4	N/C	30.4	N/C	N/C
	LoPV1	21.2	N/C	6.3	-	N/C
	MelRoV1	20.8	N/C	29.1	N/C	N/C
	PiFleV1	26.8	N/C	25.5	N/C	N/C
	RCaVV	24.3	N/C	13.7	-	N/C
	SpV1	21.4	N/C	N/C	-	N/C
PiFleV1 vs.	AAnV1	17.3	17.1	7.2	-	52.2/32.8
	AMVV1	19.1	15.3	9.5	-	51.3/32.7
	BrRV1	22.9	17.5	6.6	-	52.3/32.2
	LBVaV	23.4	16.4	25.4	14.5	51.6/32.4
	LoPV1	17.1	17.1	8.8	-	52.5/32.1
	MelRoV1	19.7	17.4	27.7	14.2	50.9/31.5
	PhPiV1	26.8	N/C	25.5	N/C	N/C
	RCaVV	20.4	15.9	6.8	-	52.0/31.1
	SpV1	20.5	15.0	N/C	-	51.7/32.4
SpV1 vs.	AAnV1	19.3	18.5	17.4	-	57.1/39.3
	AMVV1	18.3	16.6	N/C	-	53.1/35.5
	BrRV1	24.6	18.1	N/C	-	59.8/51.5
	LBVaV	18.8	16.7	N/C	-	53.5/36.1
	LoPV1	18.8	16.8	N/C	-	54.1/35.9
	MelRoV1	21.9	20.5	N/C	-	56.3/41.0
	PhPiV1	21.4	N/C	N/C	-	N/C
	PiFleV1	20.5	15.0	N/C	-	51.7/32.4
	RCaVV	27.7	18.5	N/C	-	60.8/54.8

^a^ Nucleotide percentages; ^b^ amino acid percentages; N/C: not complete; virus names are listed in Appendix A and Table 1.

## Data Availability

The sequences deposited in GenBank with accession numbers BK014297-014366 and BK059208-BK059209 are available as Appendix A.

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
