# Peer review of "Illuminating the Plant Rhabdovirus Landscape through Metatranscriptomics Data"

_viruses, 2021, doi:10.3390/v13071304_

Round 1
Reviewer 1 Report
Illuminating the plant rhabdovirus….
The manuscript presents the results of a growing trend in virology of data mining existing metagenomic datasets for the purpose of virus discovery. The focus of this article is on novel rhabdoviruses in plant and arthropod SRAs. The result is the discovery of an impressive 26 novel rhabdo-like viruses, some of which appear to possess diverse genomic properties. The manuscript is very well written, comprehensive, and furthers our knowledge on this family, and I recommend publication once my comments below are addressed:
Major comments:
Like any study that uses HTS to discover and characterize a novel virus, it is appropriate to report some relevant statistics, which at minimum should include the number/% of viral reads in the dataset and depth of coverage for each of the new finds. This request can be accommodated by additional columns in Table 1, or a new supplementary table.
The authors promote the benefits of using data mining for virus discovery, and I am full agreement with them on the benefits of this approach. However, I also realize there are concerns with this approach. Despite the efforts of its curators, submission errors exist in Genbank accessions, and this can potentially be amplified by the authors use of unpublished datasets. Were the host species correctly identified or did any of these datasets have contaminating reads? For the latter, there are always a percentage of reads in our HTS data some can be linked to either laboratory contamination, or contamination at core facilities and other sequencing companies. I have no doubt the described viruses exist, but are we certain they are linked to the actual plant (or insect) host for which the viruses are named? The inability of going back to biological material to confirm the presence of the virus, or confirm unusual genome organizations by conventional sequencing (which is a common request of many reviewers) is clearly a drawback of this approach. I would suggest the authors dedicate a small paragraph in the discussion that highlight some of the potential limitations of this data mining approach.
The layout of Figure 1 needs improvement for clarity. 1A is excellent with the single column approach. The genome depiction of the 7 viruses at the bottom of 1B should be more clearly connected the rest of the Cytorhabdoviruses. I would suggest making 1C a single column, and not indenting 1D.
Figure 2: I think it would be helpful to distinguish viruses recognized by the ICTV in this figure, perhaps by a different rectangle color. I am not familiar with rhabdovirus nomenclature, but there seems to be an inconsistency in capitalization of virus names with the host binomial (trifolium pratense virus A vs Persimmon virus A vs Spinach virus 1), as well as the use of the genus in the name (X virus vs X cytorhabdovirus). Are these virus or species names? For example, I see “red clover associated varicosavirus; is this a virus of the species Trifolium varicosavirus? Finally, there appear to be a lot of branches with very poor support (less than 50%). Is this common for this virus family? Will these genera (existing and proposed) adequately hold up if these branches are collapsed due to poor support? Perhaps a different analysis such as bayesian might be helpful.
The lone virus discovered from an arthropod SRA was from whitefly. On line 528 the authors speculate it must be a whitefly-transmitted rhabdovirus. Could it not also simply be rhabdovirus sequences ingested by the whitefly, since whiteflies don’t seem to be a traditional rhabdovirus vector? The authors should consider citing https://doi.org/10.3390/v12091028 to support their argument that whitefly could be a vector.
The authors propose Varicosavirus be split into two genera based on a deep branch in the phylogenetic tree. I see the opportunity to split several of these genera based solely on deep branching (ie Alphanucleorhabdovirus). Is there any other other basis to support this split? Personally, I find the putative segmented genome of PiFleV1 to be a more compelling reason for establishing a new varicosavirus genus.
Is the supplementary Table 3 data complete? It appears there may be missing Cytorhabdoviruses in the table.
Minor comments:
- Bemisia is spelled incorrectly ("Bemicia") throughout the document, including the phylogenetic tree
- For Allium chinense virus 1, Chinense is sometimes capitalized, which I believe is inappropriate for a binomial in a virus name
- L49: Perhaps “…the great number of viruses recently discovered by HTS…”
- L95: elsewhere “transcriptome datasets” is used, perhaps for consistency continue using this term instead of "RNA-seq" datasets
- L111: “returning rhabdovirus-like hits”
- L119-21: this sentence is a little awkward, please rephrase
- L156: I think only ICTV can assign to a genus; use propose or tentative
- L164 and 215: should these headings be italicized here, since they are genera (journal format issue)
- L167: should Blue be capitalized?
- L175: should there be a gap between the G and U?
- L188: remove extra period at end of sentence
- Table 3: how were viruses selected? Highest sequence identity? When reporting sequence identities in the results section, are only the values for the viruses “selected” for Table 3 reported, or all viruses in the genus?
- L261: Is there a reason Alpha- and Beta-nucleorhabdoviruses are addressed separately, but all “cytorhabdoviruses” are grouped?
- L317 to 333: I am having difficulty understanding the virus name formatting: should Trifolium pratense and Bacopa monnieri be italicized? Should Colocasia be either italicized or lower case? Why is Paper capitalized?
- L380: should there be spaces between U G C?
- L447: “…and SuSV1, due to incomplete genome sequence data, cannot…”
- L471-3: I don’t necessarily disagree, but what is the reasoning for this statement?
- L559: remove “tequila”
- L564,7: TaVCV
- L600: recognized by whom?
- Dichorhavirus is misspelled in Figure 2
Author Response
The manuscript presents the results of a growing trend in virology of data mining existing metagenomic datasets for the purpose of virus discovery. The focus of this article is on novel rhabdoviruses in plant and arthropod SRAs. The result is the discovery of an impressive 26 novel rhabdo-like viruses, some of which appear to possess diverse genomic properties. The manuscript is very well written, comprehensive, and furthers our knowledge on this family, and I recommend publication once my comments below are addressed.
We thank reviewer #1 for taking the time to thoroughly assess our MS and provide suggestions which improved the MS.
Major comments:
Like any study that uses HTS to discover and characterize a novel virus, it is appropriate to report some relevant statistics, which at minimum should include the number/% of viral reads in the dataset and depth of coverage for each of the new finds. This request can be accommodated by additional columns in Table 1, or a new supplementary table.
We added a Supplementary Table, named Table S2 where the relevant statistics associated with the assembly of each virus genome are shown
The authors promote the benefits of using data mining for virus discovery, and I am full agreement with them on the benefits of this approach. However, I also realize there are concerns with this approach. Despite the efforts of its curators, submission errors exist in Genbank accessions, and this can potentially be amplified by the authors use of unpublished datasets. Were the host species correctly identified or did any of these datasets have contaminating reads? For the latter, there are always a percentage of reads in our HTS data some can be linked to either laboratory contamination, or contamination at core facilities and other sequencing companies. I have no doubt the described viruses exist, but are we certain they are linked to the actual plant (or insect) host for which the viruses are named? The inability of going back to biological material to confirm the presence of the virus, or confirm unusual genome organizations by conventional sequencing (which is a common request of many reviewers) is clearly a drawback of this approach. I would suggest the authors dedicate a small paragraph in the discussion that highlight some of the potential limitations of this data mining approach.
A small paragraph that reads as: “However, the inability of going back to the biological material to confirm the viral genome sequence and to link the presence of the virus to a specific host, is the main drawback of the data mining approach for virus discovery. This limitation could lead to the potential misidentification of the host species linked to the virus. Thus, researchers need to be cautious when analyzing SRA public data for virus discovery” was added in the Concluding Remarks section.
The layout of Figure 1 needs improvement for clarity. 1A is excellent with the single column approach. The genome depiction of the 7 viruses at the bottom of 1B should be more clearly connected the rest of the Cytorhabdoviruses. I would suggest making 1C a single column, and not indenting 1D.
Figure 1 was modified accordingly
Figure 2: I think it would be helpful to distinguish viruses recognized by the ICTV in this figure, perhaps by a different rectangle color.
We think that it would be better to only distinguish those novel viruses assembled in our study.
I am not familiar with rhabdovirus nomenclature, but there seems to be an inconsistency in capitalization of virus names with the host binomial (trifolium pratense virus A vs Persimmon virus A vs Spinach virus 1), as well as the use of the genus in the name (X virus vs X cytorhabdovirus). Are these virus or species names? For example, I see “red clover associated varicosavirus; is this a virus of the species Trifolium varicosavirus?
To remove these inconsistencies, Trifolium is now capitalized, while persimmon and spinach are written in lower case. Regarding the use of the genus in the virus name, we use the same virus name as that reported by the authors who published it.
Finally, there appear to be a lot of branches with very poor support (less than 50%). Is this common for this virus family? Will these genera (existing and proposed) adequately hold up if these branches are collapsed due to poor support? Perhaps a different analysis such as bayesian might be helpful.
The poor support in some branches is quite common for plant rhabdoviruses and genera (existing and proposed) holds up if the branches with poor support are collapsed. Bayesian and ML trees yielded comparable results in our analyses.
The lone virus discovered from an arthropod SRA was from whitefly. On line 528 the authors speculate it must be a whitefly-transmitted rhabdovirus. Could it not also simply be rhabdovirus sequences ingested by the whitefly, since whiteflies don’t seem to be a traditional rhabdovirus vector? The authors should consider citing https://doi.org/10.3390/v12091028 to support their argument that whitefly could be a vector.
To support the argument that whitefly could be a cytorhabdovirus vector, we added a sentence that now reads as: “Recently, it was reported that bean-associated cytorhabdovirus is efficiently transmitted by whiteflies, which was the first report of a whitefly-transmitted rhabdovirus [64], thus supporting our hypothesis that soybean blotchy mosaic virus is likely the second such whitefly-transmitted rhabdovirus.”
The authors propose Varicosavirus be split into two genera based on a deep branch in the phylogenetic tree. I see the opportunity to split several of these genera based solely on deep branching (ie Alphanucleorhabdovirus). Is there any other basis to support this split? Personally, I find the putative segmented genome of PiFleV1 to be a more compelling reason for establishing a new varicosavirus genus.
We agree with the argument made by the reviewer and will no longer propose to split the genus Varicosavirus in the revised version of the manuscript. Thus, we deleted the last paragraph of section 4.7 in the Discussion.
Is the supplementary Table 3 data complete? It appears there may be missing Cytorhabdoviruses in the table.
Table S3 is complete. Only those viruses for which coding-complete sequences are available were included. Bean-associated cytorhabdovirus and citrus-associated rhabdovirus are strains of papaya virus E, while strawberry-associated virus 1 is an isolate of strawberry virus 1. Thus, these viruses were not included in Table S3.
Minor comments:
- Bemisia is spelled incorrectly ("Bemicia") throughout the document, including the phylogenetic tree
corrected
- For Allium chinense virus 1, Chinense is sometimes capitalized, which I believe is inappropriate for a binomial in a virus name
corrected
- L49: Perhaps “…the great number of viruses recently discovered by HTS…”
corrected
- L95: elsewhere “transcriptome datasets” is used, perhaps for consistency continue using this term instead of "RNA-seq" datasets
We replaced “RNA-seq” with “transcriptome”
- L111: “returning rhabdovirus-like hits”
corrected
- L119-21: this sentence is a little awkward, please rephrase
The sentence was rephrased, and now reads as follows: “This strategy employed BLAST/nhmmer to extract a subset of reads related to the query contig, and these retrieved reads were used to extend the contig and then the process was repeated iteratively using as query the extended sequence.”
- L156: I think only ICTV can assign to a genus; use propose or tentative
Tentative was added, so the sentence now reads: “the novel viruses were tentative assigned”
- L164 and 215: should these headings be italicized here, since they are genera (journal format issue)
We think that these headings should remain italicized
- L167: should Blue be capitalized?
corrected
- L175: should there be a gap between the G and U?
corrected
- L188: remove extra period at end of sentence
corrected
- Table 3: how were viruses selected? Highest sequence identity? When reporting sequence identities in the results section, are only the values for the viruses “selected” for Table 3 reported, or all viruses in the genus?
The viruses selected were those with the highest sequence identity. The reported sequence identities are those corresponding to the viruses “selected” for Table 3
- L261: Is there a reason Alpha- and Beta-nucleorhabdoviruses are addressed separately, but all “cytorhabdoviruses” are grouped?
The reason is that Alpha and Beta-nucleorhabdovirus are two different genera, while, at the moment, all cytorhabdoviruses belong to the one genus Cytorhabdovirus.
- L317 to 333: I am having difficulty understanding the virus name formatting: should Trifolium pratense and Bacopa monnieri be italicized? Should Colocasia be either italicized or lower case? Why is Paper capitalized?
To be consistent regarding the virus name formatting, Trifolium pratense and Bacopa monnieri have been italicized, while paper is not capitalized.
- L380: should there be spaces between U G C?
corrected
- L447: “…and SuSV1, due to incomplete genome sequence data, cannot…”
corrected
- L471-3: I don’t necessarily disagree, but what is the reasoning for this statement?
The reasoning here is that because virus discovery studies mainly have been focused on symptomatic plants, or ecological studies where hosts are dicots, the gymnosperm virome has been neglected, thus it should be further explored.
- L559: remove “tequila”
corrected
- L564,7: TaVCV
corrected
- L600: recognized by whom?
The sentence has been revised: “Currently there are only three varicosaviruses recognized by the ICTV”
- Dichorhavirus is misspelled in Figure 2
Corrected
Reviewer 2 Report
Bejerman et al. report the identification of novel virus species of family Rhabdoviridae result of the exploration of raw HTS sequences available in SRA public databases. The genetic analysis and bioinformatics tools used in the study are adequate and the analysis processes are well described. This methodology for the identification of new virus species, and even known viruses present in undescribed species, can be extended to other families of plant viruses and other hosts. In this way, this work demonstrates that mining bulk sequences in databases expands our knowledge of the virosphere.
There are several interesting findings in the manuscript, among which I highlight only the main ones. Twenty-four new species of the family Rhabbdoviridae are described, of which it has been possible to reconstruct the complete genomes of 16 and of another ten, segments partially covering the corresponding genomes have been obtained. Relevant novel features of the proteins encoded by the different viruses characterized are also described in addition to novel genomic organizations in some of the new viruses described. A very interesting discovery is that of the non-segmented genome varicosavirus PiFleV1, which extends the possibilities of genomic organization in the genus, which until now had only known members consisting of bipartite genomes. Besides, this study supports the possibility that only 4 genes are necessary for the replication of varicosaviruses and even cytorhabdoviruses.
A missing piece of information that I think could be interesting is whether the authors found rhabdoviruses already described and how often viruses of this family appear in the set of SRA sequences available in the public databases.
Comments:
L114-116: On which platform or software were the BLASTX analyses performed locally - Geneious?
L143: Why has the phylogenetic analysis been performed with the N protein (nucleocapsid)? Wouldn't it have been more convenient to do it with the L protein (replicase)?
L181-183: It would be clarifying to indicate in the caption of Figure 1 the correspondence between the colors and the functions of the proteins.
L209: I would like to see the phylogenetic tree made with the L protein to compare the topology and substitution model with that of the N protein. This would be particularly interesting, in my opinion, in the case of the varicosavirus Pinus flexilis virus 1 that plausibly consists of a monopartite genome.
L519-531: The conclusions regarding the correlation between phylogeny and transmission can be strengthened, or at least considered, if a phylogenetic tree of the L protein is also available. The same can be said in the rest of the discussion regarding the phylogenetic grouping of the newly described rhabdoviruses.
L610: This is a repetition of L367. Remove or rephrase sentence to avoid repetition in the discussion of the basic characteristics of the L protein of PiFLeV1.
L639-L642: This conclusion can only be reached if the phylogenetic tree of the replicase is available.
Author Response
Bejerman et al. report the identification of novel virus species of family Rhabdoviridae result of the exploration of raw HTS sequences available in SRA public databases. The genetic analysis and bioinformatics tools used in the study are adequate and the analysis processes are well described. This methodology for the identification of new virus species, and even known viruses present in undescribed species, can be extended to other families of plant viruses and other hosts. In this way, this work demonstrates that mining bulk sequences in databases expands our knowledge of the virosphere.
There are several interesting findings in the manuscript, among which I highlight only the main ones. Twenty-four new species of the family Rhabbdoviridae are described, of which it has been possible to reconstruct the complete genomes of 16 and of another ten, segments partially covering the corresponding genomes have been obtained. Relevant novel features of the proteins encoded by the different viruses characterized are also described in addition to novel genomic organizations in some of the new viruses described. A very interesting discovery is that of the non-segmented genome varicosavirus PiFleV1, which extends the possibilities of genomic organization in the genus, which until now had only known members consisting of bipartite genomes. Besides, this study supports the possibility that only 4 genes are necessary for the replication of varicosaviruses and even cytorhabdoviruses.
We thank reviewer #2 for taking the time to thoroughly assess our MS and provide suggestions which improved the MS.
A missing piece of information that I think could be interesting is whether the authors found rhabdoviruses already described and how often viruses of this family appear in the set of SRA sequences available in the public databases.
We found sequences of rhabdoviruses already described, including alfalfa dwarf virus (PRJNA179114, Medicago sativa), papaya virus E (PRJNA551353, Jasminum sambac; PRJNA491235, Pelargonium radula), lettuce big-vein associated virus (PRJNA417356, Cichorium endivia; PRJNA65477, Lactuca sativa), Persimmon virus A (PRJNA690635, Diospyros Kaki). However, our manuscript is focused on novel rhabdoviruses found in the SRA database to illuminate the landscape of these viruses, thus we think that is not relevant to include this information in the manuscript.
Comments:
L114-116: On which platform or software were the BLASTX analyses performed locally - Geneious?
BlastX analyses were performed using the NCBI platform
L143: Why has the phylogenetic analysis been performed with the N protein (nucleocapsid)? Wouldn't it have been more convenient to do it with the L protein (replicase)?
The phylogenetic analysis was performed using the N protein because the complete or nearly complete sequence of this protein was obtained from all identified viruses; while this was not the case for the L protein (please see Table 1). Thus, we decided to use the N protein to do the phylogenetic analysis. A recent manuscript describing a novel cytorhabdovirus in zucchini (https://doi.org/10.1016/j.virusres.2020.198095) showed that a similar topology was obtained using either N or L protein. Therefore, we assumed that the same topology would have been obtained if we use the L protein sequence. Please also see attached ML tree using the nearly complete or complete L protein aa sequence. This tree was constructed using the best fit model LG+G+F.
L181-183: It would be clarifying to indicate in the caption of Figure 1 the correspondence between the colors and the functions of the proteins.
The name/putative function of the six main encoded proteins (N, P, P3, M, G and L) is described in the legend of Figure 1. Each protein has a different colour for a better visualization of the diagram.
L209: I would like to see the phylogenetic tree made with the L protein to compare the topology and substitution model with that of the N protein. This would be particularly interesting, in my opinion, in the case of the varicosavirus Pinus flexilis virus 1 that plausibly consists of a monopartite genome.
We constructed a phylogenetic tree based on the L protein including those viruses with nearly complete or complete L protein, which is attached. The topology is similar to that obtained using the N protein sequence where Pinus flexilis virus 1 clearly grouped together with varicosaviruses.
L519-531: The conclusions regarding the correlation between phylogeny and transmission can be strengthened, or at least considered, if a phylogenetic tree of the L protein is also available. The same can be said in the rest of the discussion regarding the phylogenetic grouping of the newly described rhabdoviruses.
The topology of phylogenetic trees based on N and L proteins was similar which strengthens our conclusions.
L610: This is a repetition of L367. Remove or rephrase sentence to avoid repetition in the discussion of the basic characteristics of the L protein of PiFLeV1.
The sentence in L610 was rephrased and now reads as follows: “The N protein encoded by the unsegmented PiFLeV1 is basic; on the other hand, the N proteins encoded by the bi-segmented varicosaviruses AMVV1, BrRV1, LBVaV, LoPV1, MelRoV1, PhPiV1, RCaVV, and SpV1 are acidic.”
L639-L642: This conclusion can only be reached if the phylogenetic tree of the replicase is available.
Because the topology of the phylogenetic trees based on N and L proteins are indeed similar, we can reach this conclusion.
